# DARE-BENCH: EVALUATING MODELING AND INSTRUCTION FIDELITY OF LLMS IN DATA SCIENCE

**Fan Shu**[1,*]    **Yite Wang**[2,*†]    **Ruofan Wu**[1]    **Boyi Liu**[2]
**Zhewei Yao**[2]    **Yuxiong He**[2]    **Feng Yan**[1]
[1]University of Houston    [2]Snowflake AI Research

## ABSTRACT

The fast-growing demands in using Large Language Models (LLMs) to tackle complex multi-step data science tasks create an emergent need for accurate benchmarking. There are two major gaps in existing benchmarks: (i) the lack of standardized, process-aware evaluation that captures instruction adherence and process fidelity, and (ii) the scarcity of accurately labeled training data. To bridge these gaps, we introduce DARE-bench, a benchmark designed for machine learning modeling and data science instruction following. Unlike many existing benchmarks that rely on human- or model-based judges, all tasks in DARE-bench have verifiable ground truth, ensuring objective and reproducible evaluation. To cover a broad range of tasks and support agentic tools, DARE-bench consists of 6,300 Kaggle-derived tasks and provides both large-scale training data and evaluation sets. Extensive evaluations show that even highly capable models such as gpt-o4-mini struggle to achieve good performance, especially in machine learning modeling tasks. Using DARE-bench training tasks for fine-tuning can substantially improve model performance. For example, supervised fine-tuning boosts Qwen3-32B's accuracy by $1.83\times$ and reinforcement learning boosts Qwen3-4B's accuracy by more than $8\times$. These significant improvements verify the importance of DARE-bench both as an accurate evaluation benchmark and critical training data. Our data will be released at https://github.com/Snowflake-Labs/dare-bench.

## 1 INTRODUCTION

Large language models (LLMs) (Anthropic, 2025a;b; OpenAI, 2025a;c; Yang et al., 2025) are increasingly employed as data-science (DS) agents to perform data reading, transformation, and modeling through tool-augmented code execution. Such a rapid adoption demands rigorous benchmarks to evaluate and enhance the effectiveness and reliability in performing these complex, multi-step workflows. However, due to the cost and complexity of evaluation, existing benchmarks can only evaluate final-answer accuracy, and leaving other valuable metrics such as process fidelity and reproducibility largely unmeasured (Zhang et al., 2024; Jing et al., 2024). Meanwhile, many existing works (Guo et al., 2024; Zhang et al., 2023; Hong et al., 2024) in this area focus on using prompt engineering and workflow design to improve model performance.

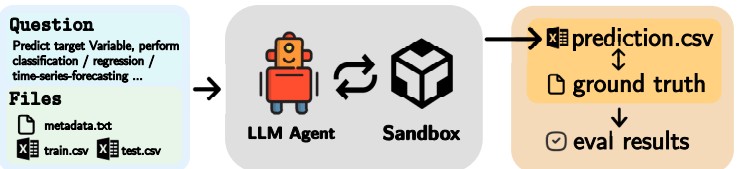

Figure 1: DARE-bench defines each task by providing a natural-language question and structured files (metadata and train/test splits). An LLM agent executes code within a sandbox to generate predictions, which are compared against ground truth for automatic and reproducible evaluation.

---

*Equal contribution. †Project lead. Work done during Fan Shu's internship at Snowflake AI Research.

Table 1: Comparison between DARE-bench and existing benchmarks.

| Benchmark | Domain | Data File | Inst-follow | Time Series | Verifiable | Train Tasks | Tasks |
|---|---|---|---|---|---|---|---|
| MLAgentBench (Huang et al., 2023) | Deep Learning | ✓ | - | - | ✓ | ✗ | 13 |
| MLE-bench (Chan et al., 2024) | Deep Learning | ✓ | - | - | ✓ | ✗ | 75 |
| SWE-bench (Jimenez et al., 2024) | Software Eng. | ✓ | - | - | ✓ | ✓ | 21,294 |
| DS-1000 (Lai et al., 2023) | Data Science | ✗ | ✗ | ✗ | ✗ | ✗ | 1,000 |
| Arcade (Yin et al., 2022) | Data Science | ✗ | ✗ | ✗ | ✗ | ✗ | 1,082 |
| Spider2V (Cao et al., 2024) | Data Science | ✗ | ✗ | ✗ | ✓ | ✗ | 494 |
| DSEval (Zhang et al., 2024) | Data Science | ✓ | ✗ | ✗ | ✓ | ✗ | 825 |
| DSBench (Jing et al., 2024) | Data Science | ✓ | ✗ | ✗ | ✓ | ✗ | 540 |
| DA-Code (Huang et al., 2024) | Data Science | ✓ | ✗ | ✗ | ✓ | ✗ | 500 |
| DataSciBench (Zhang et al., 2025) | Data Science | ✓ | ✗ | ✗ | ✗ | ✗ | 222 |
| DABstep (Egg et al., 2025a) | Data Science | ✓ | ✗ | ✗ | ✓ | ✗ | 450 |
| DSBC (Kadiyala et al., 2025) | Data Science | ✓ | ✗ | ✗ | ✓ | ✗ | 303 |
| DARE-bench (Ours) | Data Science | ✓ | ✓ | ✓ | ✓ | ✓ | 6,300 |

We complement these works by taking a benchmark approach to train LLM agents with high fidelity data and sophisticated yet reproducible evaluation to better acquire domain-specific skills in DS workflows.

Creating benchmarks that capture process fidelity for both training and evaluation is significantly challenging. The main challenge comes from two-fold. First, the sources for crafting training data (e.g., expert-level, executable DS process traces) are scarce and prohibitively expensive to acquire. Existing benchmarks largely rely on human-processed data and often center on Kaggle competitions, creating a major data bottleneck. Second, evaluating "process fidelity" is highly non-trivial as randomness and environmental effects confound behavior, and verifying that an agent follows permissible DS practices requires a controlled, instrumented harness. These challenges limit the data quality and evaluation scope of existing benchmarks, and thus miss the opportunities to better release the full potential of models.

To address the challenge of data quality and scarcity, we leverage LLMs to process auxiliary content, such as task descriptions, metadata normalization, rule extraction, instead of heavily relying on human involvement so that the data generation is scalable with quality. We further improve the data quality with better diversity by pivoting from leaderboard-oriented Kaggle competitions to the broader pool of Kaggle datasets, yielding a more diverse and representative problem set such as time-series domains. To address the evaluation challenge, we engineer determinism (e.g., fixed seeds, reproducible environments) so that process fidelity is enabled by an outcome-based, verifiable reward—enabling reinforcement learning (RLVR) instead of human-involved reward. These approaches work coherently to construct a large-scale, trainable benchmark for data science that measures modeling performance and process fidelity, and boosts training performance.

To this end, we introduce **D**atascience **A**gentic **RE**asoning bench (DARE-bench), a training-focused DS agent benchmark featuring two verifiable task families: (i) process-aware instruction-following tasks with ground truth from executing reference solutions that strictly follow the task instruction; and (ii) ML modeling tasks evaluated against the dataset's original ground truth under reproducible metrics. Our design for the instruction-following tasks leverages a key advantage of data science: the high degree of reproducibility. We find that by controlling the randomness and providing explicit instructions, a procedurally faithful execution can produce a deterministic outcome. This allows us to robustly and automatically evaluate process fidelity by verifying the agent's final answer against the ground truth. As shown in Figure 1, for both task families, each task provides a natural-language question and structured files. The LLMs execute code within a sandbox to generate predictions, which is checked automatically for scoring. In Table 1, we compare DARE-bench against existing benchmarks in terms of the task coverage, verifiability, training task support, and number of tasks to demonstrate DARE-bench's significant advancements.

We conduct extensive evaluation on both strong general-purpose and code-centric LLMs. The evaluation results reveal that many LLMs without task-aligned training fail miserably due to process deviations, runtime errors, and metric mis-specification. For instance, Qwen3-32B baseline only

Table 2: Overview of DARE-bench benchmark composition and the primary capabilities evaluated by each task type. Variants are denoted as IF = Instruction Following, MM = ML Modeling, XF = eXogenous Features, CF = Canonical Forecasting.

| Task Type | Train Tasks | Test Tasks | Capability Assessed |
|---|---|---|---|
| Classification-IF | 1160 | 74 | Instruction following |
| Classification-MM | 1160 | 74 | ML Modeling |
| Regression-IF | 899 | 45 | Instruction following |
| Regression-MM | 899 | 45 | ML Modeling |
| Time-series-XF | 915 | 57 | Predictive ML, forecasting |
| Time-series-CF | 915 | 57 | Predictive ML, forecasting |

achieves a total score of 23.25, while the smaller Qwen3-4B baseline performs even worse which scores 4.39. By contrast, DARE-bench bridges this gap by providing a training-focused benchmark with verifiable large-scale training data and useful and sophisticated reproducible evaluation. Supervised fine-tuning yields absolute gains of nearly 20 points, while reinforcement learning boosts Qwen3-4B from 4.39 to 37.40. Overall, DARE-bench significantly improve success rates, process adherence, predictive performance, and robustness across a variety of practical data science tasks.

## 2 RELATED WORK

**LLM Agents.** Research into Agentic LLMs focuses on their ability as independent agents through planning, tool calling, and memory capabilities. The integration of reasoning with actions or APIs occurs through ReAct (Yao et al., 2023) and Toolformer (Schick et al., 2023) frameworks as researchers work on multi-agent collaboration and autonomous tool-augmented systems. Applying these to real-world data science remains difficult because current benchmarks lack adequate training resources and often omit critical domains such as time-series forecasting or the distinction between open-ended problem solving and strict instruction-following.

**LLMs for Coding and Data Science Benchmarks.** The advancement of coding benchmarks depends on the use of testable pass/fail signals. The HumanEval (Chen et al., 2021) and MBPP (Austin et al., 2021) provided short self-contained functions with hidden unit tests while SWE-bench (Jimenez et al., 2024) tests models on actual GitHub issues that need multiple file modifications and complete project testing. The community now performs end-to-end data science (DS) tasks as its new approach to this paradigm. The DS-1000 (Lai et al., 2023) teaches NumPy/Pandas programming but DSBench (Jing et al., 2024) and MLE-bench (Chan et al., 2024) use Kaggle competition problems which require multi-step analytics. The DABstep (Egg et al., 2025b) dataset contains 450 financial tasks from real-world applications and DataSciBench (Zhang et al., 2025) uses Task-Function-Code (TFC) to evaluate programs which are then verified by human evaluators. DSBC (Kadiyala et al., 2025) addresses private datasets via structured metadata. The research uses Chen et al. (2024) to evaluate visualization skills and Bendinelli et al. (2025) to assess data cleaning abilities and Kaggle leaderboards (Grosnit et al., 2024; Chan et al., 2024) to measure performance. The benchmarks show a sequential development from basic unit testing code to sophisticated tool-based agents which perform complete DS workflows and produce quantifiable results.

**Reinforcement Learning with Verifiable Rewards.** The implementation of verifiable programmatic signals in reinforcement learning enables model training at scale without requiring preference data. The automatic checking system consists of unit tests and solvers and execution traces for math and code verification. GRPO (Shao et al., 2024) achieves learning stability through its relative rollout feedback system which DeepSeek-R1 (Guo et al., 2025) and GPT o-series (OpenAI, 2025d) extend by verifier-enhanced objectives. The methods combine symbolic proofs with coding tests and retrieval/search execution graphs to improve reward-as-checker for both correct answers and verifiable reasoning trace generation.

# 3 DARE-BENCH

DARE-bench consists of three data science task-families - classification, regression and time-series forecasting, each with two variants that probe distinct agent capabilities. For clarity, we denote these variants using intuitive abbreviations: **IF** (Instruction Following) and **MM** (ML Modeling) for classification and regression; **XF** (eXogenous Features) and **CF** (Canonical Forecasting) for time-series forecasting. In classification and regression, the IF variant emphasizes instruction-following by requiring LLM to faithfully reproduce reference workflows, whereas the MM variant targets ML modeling with outcome-based evaluation. These variants capture complementary real-world needs. IF simulates a workflow where an agent must strictly execute a senior scientist's detailed design. Conversely, MM reflects an outcome-driven scenario where customers only care about the final accuracy, granting full freedom to the LLM. For time-series forecasting, the distinction between the two variants is more nuanced: in the XF variant, we retain not only the timestamp and entity identification columns but also all exogenous features from the original dataset; in the CF variant, however, while exogenous features remain available for training, the test set is constrained to only the timestamp and entity columns, making it closer to a classical forecasting setup. We partition our collection of 6,300 tasks into an approximately 95/5 train/test split, designating the most recently updated tasks as the test set. Table 2 summarizes the dataset scale and the primary capability assessed in each task type. Tool schema and task examples are shown in Appendix K.

## 3.1 DATASET CURATION

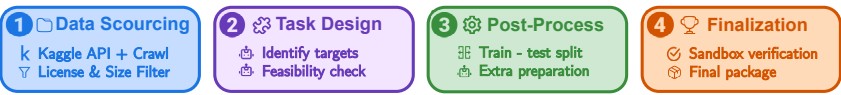

Figure 2: **Automated pipeline of DARE-bench.** The construction process consists of four stages: (1) *Dataset Sourcing*, where Kaggle datasets are filtered by tags, license, size, and metadata; (2) *Task Design*, where schema summaries, targets, features, and feasibility are analyzed with the help of LLM; (3) *Post-Process*, including splitting, noise injection for IF tasks or resampling or entity checks for time-series-CF tasks; and (4) *Finalization*, which validates solvability in a sandbox for IF tasks and produces standardized benchmark artifacts.

To construct DARE-bench, we design an automated data curation pipeline that systematically transforms raw Kaggle datasets into standardized machine learning tasks. Unlike prior benchmarks which rely mainly on manual curation, our approach integrates web crawling, LLM-based task formulation, controlled data transformations, and sandbox verification to ensure both quality and scale. Shown in Figure 2, the pipeline consists of four stages. Detailed prompts are shown in Appendix I.

**Dataset Sourcing with Augmented Metadata.** We selected Kaggle as the primary data source due to its breadth of real-world, user-contributed datasets. The official API of Kaggle retrieves candidate datasets that meet specific criteria including tabular format and valid open license. Additionally, we develop a lightweight web crawler to extract additional data from webpage descriptions that were present in the dataset, providing additional metadata elements to the LLM through column previews and natural-language descriptions which help the model understand the context of the task formulation.

**LLM-Assisted Task Design and Feasibility Analysis.** For each sourced candidate dataset, we employ an LLM to assess whether it can support a well-posed predictive task. The model receives both the dataset preview and the detailed description to duplicate expert assessment on a large scale. The LLM detects a target column which can be either categorical or continuous for classification and regression tasks along with structured features and their corresponding data types. For time-series forecasting tasks, the model detects timestamp columns and numerical targets that evolve through time and exogenous features in addition to identifying the temporal frequency of the data. Only datasets deemed feasible by this automated analysis proceed to the next stage.

**Post-Process.** Feasible datasets are then transformed into uniform benchmarking tasks. The data is split randomly into training and testing sets. For instruction-following tasks, controlled noise is injected into roughly twenty percent of the training data, which simulates real-world data quality

issues through numerical values that exceed valid ranges and unexpected categorical entries, and the testing set serves as the clean reference data. The chronological split method is used for time-series forecasting to preserve the natural order of time in the data. LLM then detects entity identifiers to stop data leakage between groups and it performs automatic resampling of irregular time series data to uniform intervals through an aggregation method suggested by the model.

**Finalization.** After the post-process step, for instruction-following tasks, the validation process for each task runs independently in a sandbox environment by executing the reference solution code sequence including data loading, preprocessing, training, and prediction generation. Since these tasks rely on reference outputs rather than fixed ground truth values, the sandbox ensures that the instructions can be faithfully executed and the generated predictions are fully reproducible under the same random seed. In contrast, ML modeling tasks directly use ground-truth values (e.g., class labels or numerical targets) for evaluation and do not require sandbox execution. Finally, the task is packaged into a standardized format that includes training and testing data, metadata describing the dataset and task, the natural language task description, and the corresponding reference.

## 3.2 Task Formulation

**Input and output.** Suppose we have the task description $Q$, an accompanying dataset description $M$, a training set $\mathcal{D}_{\text{train}} = \{(\mathbf{x}_i, \mathbf{y}_i)\}_{i=1}^{n_{\text{train}}}$, a testing set without target values $\mathcal{D}_{\text{test}} = \{\mathbf{x}_i\}_{i=1}^{n_{\text{test}}}$, and access to a code execution tool $\mathcal{T}$. The tool $\mathcal{T}$ enforces a maximum wall-clock runtime $T_{\text{max}}$, while the agent $\mathcal{G}$ is subject to an interaction budget of $K$ turns. Given these inputs and constraints, $\mathcal{G}$ produces executable code $\mathcal{C}$, which is run within $\mathcal{T}$ on $\mathcal{D}_{\text{train}}$ to fit a model and subsequently on $\mathcal{D}_{\text{test}}$ to generate predictions $\hat{\mathbf{y}}$, i.e., $\hat{\mathbf{y}} = \mathcal{G}(Q, \mathcal{D}_{\text{train}}, \mathcal{D}_{\text{test}}, M, \mathcal{T}(T_{\text{max}}, K))$.

**Evaluation metrics.** We evaluate models differently depending on the task type. For instruction-following tasks (i.e., Classification-IF and Regression-IF), we compare the model's generated prediction $\hat{\mathbf{y}}$ against the simulated reference output $\mathbf{y}_{\text{ref}}$ obtained from the reference solution code $\mathcal{C}_{\text{ref}}$, and assign a score of $1$ if $\hat{\mathbf{y}} = \mathbf{y}_{\text{ref}}$ and $0$ otherwise. For ML modeling tasks, including Classification-MM, Regression-MM, and both Time-series-XF and Time-series-CF, we directly compare the model predictions $\hat{\mathbf{y}}$ against the masked ground-truth values $\mathbf{y}_{\text{gt}}$. Specifically, we adopt the macro-F1 score for classification-MM tasks to account for class imbalance, and use the clipped coefficient of determination for regression and time-series forecasting, defined as $\text{clip}(R^2) = \min\{1, \max\{0, R^2\}\}$. For tasks with multiple prediction targets, the evaluation metric is computed by averaging over all targets. Details of our reference solution code can be found in Appendix J and calculation of macro-F1 and $R^2$ can be found in Appendix D.

## 3.3 Features of DARE-bench

DARE-bench introduces several key features that distinguish it from prior benchmarks in data science and machine learning:

**ML Modeling and Instruction Following.** DARE-bench differs from other existing benchmarks because it assesses two fundamental data science capabilities which are essential for real-world applications: ML modeling and task instruction following for data processing and model development.

**Verifiable Ground Truth.** The evaluation process of DARE-bench depends on actual labels and simulated reference solution outputs to produce results that can be replicated. The system design removes all dependencies on human judgment and model-based assessments that enables evaluation metrics to directly assess task performance. This design is similar to coding benchmarks such as SWE-bench (Jimenez et al., 2024) and math benchmarks like AIME (Balunović et al., 2025), making it extremely suitable for supervised fine-tuning (SFT) and reinforcement learning with verifiable rewards (RLVR).

**Dual Role as Evaluation and Training Resource.** The benchmark offers a training dataset which enables users to perform model fine-tuning and alignment. As we will demonstrate in Section 5, the models trained on DARE-bench achieve better results than their baselines, which proves that the dataset serves as a benchmark and a resource to improve data science LLMs.

**Diversity, Realism, and Practical Constraints.** Our datasets are created from Kaggle sources, making them naturally diverse, multilingual, and spanning various domains while capturing real-

Table 3: Distribution of task domains across the DARE-bench train and test sets.

| Dataset | Finance | Health | Business | Technology | Automotive | Education | Environment | Others |
|---------|---------|--------|----------|------------|------------|-----------|-------------|--------|
| Train   | 16.9%   | 10.2%  | 7.3%     | 4.0%       | 4.5%       | 2.8%      | 6.8%        | 47.5%  |
| Test    | 17.1%   | 8.4%   | 8.2%     | 5.6%       | 3.3%       | 3.1%      | 2.4%        | 51.9%  |

Table 4: Hyperparameter sensitivity analysis for o4-mini across different turns and sandbox maximum execution time limit configurations.

| turns | time | class-IF | class-MM | reg-IF | reg-MM | time-XF | time-CF |
|-------|------|----------|----------|--------|--------|---------|---------|
| 3     | 300  | 37.16    | 55.44    | 29.71  | 51.69  | 37.99   | 6.67    |
| 5     | 200  | 67.56    | 57.89    | 53.62  | 57.60  | 42.29   | 9.67    |
| 6     | 180  | 73.42    | 61.07    | 63.76  | 60.92  | 41.59   | 9.79    |
| 8     | 120  | 73.87    | 61.42    | 65.21  | 61.05  | 42.11   | 8.82    |
| 10    | 100  | 75.22    | 63.36    | 62.31  | 62.07  | 42.03   | 10.97   |
| 15    | 100  | 76.80    | 65.88    | 66.66  | 62.41  | 40.03   | 9.92    |

world challenges such as class imbalance, missing values, and noise. As illustrated in Table 3, quantitative analysis confirms this broad coverage, showing that DARE-bench spans a wide spectrum of real-world verticals across both training and test sets. Details in categorization can be found in Appendix M. In addition, enforced constraints—such as a 10-minute execution limit and bounded tool invocation turns—reflect realistic user expectations for efficient, accurate solutions. See more details on Appendix H.

## 4 EVALUATION

In this section, we present the experimental results and analysis of several LLMs evaluated using DARE-bench.

### 4.1 EXPERIMENT SETTINGS

We experiment with state-of-the-art LLMs from open-source ones such as Qwen3-32B and Qwen3-4B (Yang et al., 2025), to proprietary models such as gpt-o4-mini (OpenAI, 2025d), gpt-4o, gpt-4.1 (OpenAI, 2025a), gpt-5 (OpenAI, 2025b), Claude-Sonnet-3.7 (Anthropic, 2025a), and Claude-Sonnet-4 (Anthropic, 2025b).

For all the experiments, we employ a greedy decoding strategy whenever applicable, along with sandbox (ByteDance-Seed Foundation Code Team, 2024) for code execution. To reduce randomness, each task is repeated three times and we report the average score. We evaluate all tasks using either accuracy or the macro-F1/clipped $R^2$ score. The **'classification-IF'** and **'regression-IF'** metrics are measured using a strict, binary (0/1) accuracy. **'classification-MM'** is measured using a graded (0.0-1.0) macro-F1 score. The remaining metrics, **'regression-MM'**, **'time-series-XF'**, and **'time-series-CF'**, are all evaluated using the clipped $R^2$ score.

We conduct our evaluation in two stages. First, we perform a sensitivity analysis on the key hyperparameters for our evaluation framework using one of the most advanced models, gpt-o4-mini, specifically turns and sandbox maximum execution time. These limits are set to simulate real-world applications, as a user would not wait infinite time for an agent to complete a task. Our goal is to find a balanced configuration. Second, with this configuration, we conduct a comprehensive comparison of several leading LLMs on our benchmark.

### 4.2 HYPERPARAMETER SENSITIVITY ANALYSIS

The results, shown in Table 4, clearly indicate a clear trend emerges: performance generally improves with a higher number of interactive turns. We observe a dramatic leap in performance when moving from 3-turn configurations to 5-turn configurations. For example, the classification-IF score jumps from 37.16 (at 3 turns, 300 s) to 67.56 (at 5 turns, 200 s). This suggests that allowing the agent more opportunities to iterate and refine its approach is crucial.

Table 5: Main evaluation results on our benchmark (test tasks) under the configuration where turns set as 5 and sandbox maximum execution time set as 200 s. The best score in each column is bolded.[2]

| Model | class-IF | class-MM | reg-IF | reg-MM | time-XF | time-CF |
|---|---|---|---|---|---|---|
| gpt-4o | 32.88 | 40.45 | 20.28 | 40.60 | 35.54 | 4.77 |
| gpt-4.1 | 55.82 | 57.83 | 52.17 | 58.62 | 40.78 | 6.60 |
| gpt-5 | **69.81** | 43.40 | **57.24** | 56.29 | 36.83 | 10.13 |
| gpt-o4-mini | 67.56 | 57.89 | 53.62 | 57.60 | 42.29 | 9.67 |
| Claude-Sonnet-3.7 | 61.48 | **61.03** | 46.37 | **63.20** | **49.88** | **13.70** |
| Claude-Sonnet-4 | 16.21 | 18.27 | 15.21 | 11.33 | 4.80 | 0.01 |
| Qwen3-32B | 17.11 | 30.71 | 15.21 | 35.86 | 26.96 | 0.00 |
| Qwen3-4B | 3.60 | 5.23 | 0.72 | 3.29 | 6.97 | 0.00 |

The highest performance on classification-IF (76.80) was achieved at the (15 turns, 100 s) setting. However, for our main model comparison, we sought a balance between performance and computational efficiency (i.e., cost and latency). We selected the (5 turns, 200 s) configuration as our standard setting. This configuration (5 turns, 200 s) serves as a robust and practical baseline; it significantly outperforms 3-turn setups and achieves strong, representative scores across all metrics (e.g., 67.56 on classification-IF, 53.62 on regression-IF, and 42.29 on time-series-XF) within a practical time constraint, i.e., approximately 1000 s user wait time in total.

## 4.3 MODEL COMPARISON

Based on our sensitivity analysis, we adopt the (5 turns, 200 s) configuration for a comprehensive comparison of all models. The main results are presented in Table 5. Statistics on the average token usage and the number of tool invocations are listed in Appendix K.

In this standardized setting, Claude-Sonnet-3.7 emerges as the top-performing model. It achieves the highest scores on four of the six evaluation metrics: 'classification-IF' (69.81), 'classification-MM' (61.03), 'regression-MM' (63.20), 'time-series-XF' (49.88) and 'time-series-CF' (13.70), demonstrating its strong overall capabilities for this benchmark. gpt-5 leads the two IF columns, achieving the highest 'classification-IF' (69.81) and 'regression-IF' (57.24).

The results also reveal marked disparities between model generations. Claude-Sonnet-4 underperforms significantly compared to its predecessor Claude-Sonnet-3.7, with notably weaker scores across all metrics. A key reason is that Claude-Sonnet-4 tends to decompose tasks into very fine-grained substeps, executing almost every small operation separately. As a result, completing a single benchmark task often requires a very large number of steps, and the model nearly always exceeds the allowed step limit, leading to premature failures. Meanwhile, among the open-source models, Qwen3-32B and Qwen3-4B perform far below the proprietary models, struggling in all categories and failing entirely on time-series-CF. This highlights that complex, multi-step data analysis in sandboxed environments remains a considerable challenge for current open-source LLMs.

Moving beyond the quantitative scores in Table 5 to understand why models fail on our benchmark, we conducted a systematic qualitative analysis of failed trajectories. Our goal is to identify the primary bottlenecks and limitations of current SOTA agents.

**Incorrect Tool Argument Passing.** A fundamental failure mode observed was that LLMs often failed to correctly interface with the code-execution tool. While the generated Python code was logically correct, they frequently mismatched tool parameters (e.g., forgetting to pass filenames), causing execution to fail before code could run. Definition of our tool can be found in Appendix K.

**Instruction Following Failures.** LLMs often ignored explicit constraints: processing steps in the wrong order, skipping required transformations, or omitting critical function arguments (Figure 3). These errors show weak adherence to task specifications.

**Flawed Reasoning in Open-Ended Tasks.** More subtle problems came from brittle reasoning. Common issues included misuse of metadata (hard-coding values), risky preprocessing (e.g., naive

---

[2]Due to licensing restrictions on specific source datasets, this table reports the performance on the full benchmark. We provide the performance on the strictly open-sourceable subset in Appendix E.

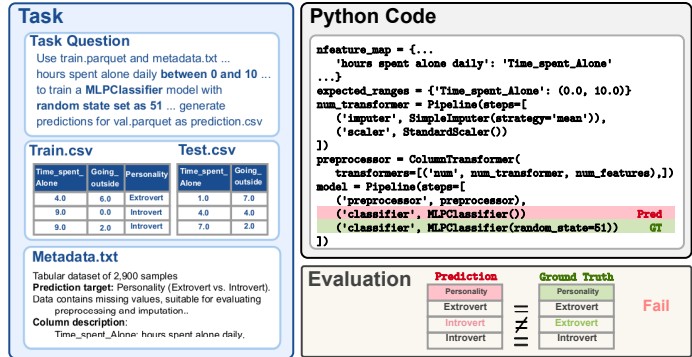

Figure 3: Example of an instruction-following task where the agent fails to respect explicit constraints. Despite being asked to fix the random seed, the model omitted the required argument, leading to incorrect predictions and an evaluation failure.

Table 6: Fine-tuning and RL improve performance over baselines. Superscripts denote absolute gains compared to the baseline of the same model.

| Model | Setting | class-IF | class-MM | reg-IF | reg-MM | time-XF | time-CF | Total | Model-Perf |
|---|---|---|---|---|---|---|---|---|---|
| Qwen3-32B | Baseline | 17.11 | 30.71 | 15.21 | 35.86 | 26.96 | 0.00 | 23.25 | 65.03 |
| Qwen3-32B | SFT-FV | $40.54^{+23.43}$ | $44.71^{+13.99}$ | $42.75^{+27.54}$ | $49.21^{+13.35}$ | $39.95^{+12.99}$ | $0.07^{+0.07}$ | $42.42^{+19.17}$ | $72.32^{+7.29}$ |
| Qwen3-32B | SFT-AV | $40.54^{+23.43}$ | $47.20^{+16.49}$ | $42.02^{+26.81}$ | $55.56^{+19.70}$ | $33.56^{+6.60}$ | $0.00^{+0.00}$ | $42.91^{+19.72}$ | $70.27^{+5.24}$ |
| Qwen3-32B | SFT-BV | $38.06^{+20.95}$ | $48.91^{+18.20}$ | $42.75^{+27.54}$ | $54.55^{+18.69}$ | $35.91^{+8.95}$ | $0.00^{+0.00}$ | $42.83^{+19.58}$ | $71.01^{+5.98}$ |
| Qwen3-32B | SFT-DV | $38.58^{+21.47}$ | $43.82^{+13.11}$ | $39.13^{+23.92}$ | $51.00^{+15.14}$ | $38.92^{+11.96}$ | $0.00^{+0.00}$ | $41.12^{+17.18}$ | $71.68^{+6.65}$ |
| Qwen3-4B | Baseline | 3.60 | 5.23 | 0.72 | 3.29 | 6.97 | 0.00 | 4.39 | 54.18 |
| Qwen3-4B | RL | $38.96^{+35.36}$ | $39.44^{+34.21}$ | $31.88^{+31.16}$ | $37.04^{+33.75}$ | $32.28^{+25.31}$ | $2.28^{+2.28}$ | $37.40^{+33.01}$ | $62.55^{+8.37}$ |

label encoding, mishandling NaNs), and unreliable type inference. Such shortcuts led to fragile pipelines and frequent errors.

**Time-Series Task Failures.** Performance on 'time-series-CF' was especially poor. Reflecting a lack of exposure to complex time-series reasoning, LLMs often failed to produce valid output formats or relied on trivial heuristics (last value, mean), resulting in near-zero predictive accuracy.

This qualitative analysis reveals that current agent failures are multi-faceted. They range from basic API misuse and poor instruction following to, most critically, a lack of robust, generalizable reasoning for complex tasks. The widespread use of brittle preprocessing and the near-total failure on complex `time-series` formatting suggest that current agents, while proficient at simple code generation, still lack the deep, domain-specific reasoning required for autonomous data science.

# 5    FINE-TUNING LLMS WITH DARE-BENCH

To further strengthen the performance of foundation LLMs on DARE-bench, we explore two complementary training paradigms: supervised fine-tuning (SFT) and reinforcement learning (RL). SFT leverages curated supervision from rejection-sampled traces to align models more closely with task requirements, while RL directly optimizes models with verifiable outcome rewards. The following subsections detail each approach, their implementation, and the improvements they yield.

**Rejection Sampling and Supervised Fine-tuning.** To obtain high-quality supervision signals, we rejection-sample traces generated across multiple runs, using task-specific filtering strategies.

We generate data for supervised fine-tuning through rejection sampling using task-independent filters that evaluate trajectories for *validity*, *quality*, and *speed*. A trajectory is *valid* if it achieves exact match for IF tasks or exceeds a type-specific score threshold for predictive tasks. A task is considered *diverse* if its sampled runs contain both successes and failures (IF) or if the variance of its

Table 7: Ablation study on the impact of Instruction Following (IF) and ML Modeling (MM) data with SFT-DV rejection sampling data.

| Train Data | class-IF | class-MM | reg-IF | reg-MM | time-XF | time-CF |
|---|---|---|---|---|---|---|
| baseline | 17.11 | 30.71 | 15.21 | 35.86 | 26.96 | 0.00 |
| IF | $40.99^{+23.88}$ | $22.38^{-8.33}$ | $47.82^{+32.61}$ | $27.85^{-8.01}$ | $23.83^{-3.13}$ | $0.00^{+0.00}$ |
| MM | $11.71^{-5.40}$ | $45.69^{+14.98}$ | $18.84^{+3.63}$ | $45.38^{+9.52}$ | $34.12^{+7.16}$ | $0.00^{+0.00}$ |
| IF+MM | $38.58^{+21.47}$ | $43.82^{+13.11}$ | $39.13^{+23.92}$ | $51.00^{+15.14}$ | $38.92^{+11.96}$ | $0.00^{+0.00}$ |

Table 8: External validation on DSBench (Jing et al., 2024) after converting tasks into our format. Superscripts denote absolute gains over the baseline of the same model.

| Model | Setting | Competition-level Accuracy |
|---|---|---|
| Qwen3-32B | Baseline | 32.38 |
| Qwen3-32B | SFT-FV | $37.82^{+5.44}$ |
| Qwen3-32B | SFT-AV | $41.08^{+8.70}$ |
| Qwen3-32B | SFT-BV | $40.06^{+7.68}$ |
| Qwen3-32B | SFT-DV | $42.41^{+10.03}$ |
| Qwen3-4B | Baseline | 18.23 |
| Qwen3-4B | RL | $40.00^{+21.77}$ |

predictive scores exceeds a threshold. We study four strategies: **FV** (Fastest-Valid), which keeps the quickest valid trace for each task; **AV** (All-Valid), which retains all valid traces; **BV** (Best-Valid), which for diverse tasks selects the single best valid trace; and **DV** (Duo-Valid), which for diverse tasks retains the top-2 valid traces (fastest for IF, highest-scoring above the mean for predictive). Both IF and predictive tasks use their natural evaluation metrics (exact match or macro-F1 / clipped $R^2$) to define validity and rank trajectories. More details are provided in Appendix L.

**Reinforcement Learning.** We perform reinforcement learning with GRPO (Shao et al., 2024) on Qwen3-4B (Yang et al., 2025) using the DARE-Bench training tasks with the verl (Sheng et al., 2025) framework. During training, we found that the group normalization used in GRPO introduces training instability. Therefore, we chose to remove the normalization component similar to Dr.GRPO (Liu et al., 2025), which mitigates the training stability issue. Moreover, we use sequence-level aggregation as in the original GRPO, rather that token-level aggregation used by DAPO (Yu et al., 2025). Additional training details can be found in Appendix G.

**Fine-tuning Results.** Table 6 summarizes results for both SFT (Qwen3-32B) and RL (Qwen3-4B). Specifically, Model-Perf measures the quality of the model's predictions by focusing solely on successful attempts for MM tasks. This metric isolates the quality dimension from the validity dimension, confirming that fine-tuning improves the model's actual data science proficiency, not just its adherence to syntax rules. Across IF and MM tasks, fine-tuning yields substantial improvements over the baseline, with absolute gains of nearly $1.83\times$ in total score and 10% in ModelPerf. Different strategies bring complementary benefits: AV yields the strongest overall improvements for MM tasks, while FV favors IF tasks. Reinforcement learning on Qwen3-4B provides even larger relative gains, boosting the total score from 4.39 to 37.40 and ModelPerf from 54.18 to 62.55. These results confirm that DARE-bench not only improves instruction following but also translates into better downstream modeling accuracy once correct predictions are generated.

**Impact of Data Composition.** As shown in Table 7, we use SFT-DV to further investigate the specific contributions of IF and MM data through an ablation study. Training exclusively on MM data boosts predictive modeling performance but degrades instruction adherence, while training solely on IF data shows the inverse. Only the combined approach successfully integrates both capabilities, achieving a robust balance. This confirms that process-oriented and outcome-oriented tasks are complementary and essential for a comprehensive data science agent.

**Failure Analysis.** Shown in Table 9, we categorized incorrect trajectories to identify specific reasoning bottlenecks. Proprietary models mainly face problems with Code Errors, while open-source baselines frequently exceed execution limits because of inefficient exploration. Training on DARE-bench effectively mitigates these issues; notably, RL on Qwen3-4B reduces code errors by 48 percent

Table 9: Failure mode analysis across different models.

| Model | Inst Adhere | Code Error | Code Exec Limit | Max Token Limit |
|---|---|---|---|---|
| gpt-5 | 126 | 333 | 0 | 0 |
| Claude-Sonnet-3.7 | 158 | 262 | 0 | 0 |
| Qwen3-32B | 48 | 106 | 257 | 372 |
| Qwen3-32B + SFT-DV | 43 | 80 | 236 | 256 |
| Qwen3-4B | 79 | 174 | 661 | 102 |
| Qwen3-4B + RL | 49 | 91 | 331 | 119 |

Table 10: Comparison between Native Function Call[4] and DataWiseAgent on DARE-bench and DSBench.

| Framework | Model | class-IF | class-MM | reg-IF | reg-MM | time-XF | time-CF | DSBench |
|---|---|---|---|---|---|---|---|---|
| Native Function Call | Qwen3-32B | 17.11 | 30.71 | 15.21 | 35.86 | 26.96 | 0.0 | 32.38 |
| Native Function Call | Qwen3-32B + SFT-DV | 38.58 | 43.82 | 39.13 | 51.00 | 38.92 | 0.0 | 42.41 |
| DataWiseAgent | Qwen3-32B | 21.62 | 29.63 | 34.78 | 34.40 | 30.45 | 0.0 | 29.17 |

and halves code execution limit errors, demonstrating that our supervision significantly enhances both code correctness and efficiency.

**Case Studies of Fine-tuning Effects.** To further illustrate the benefits of fine-tuning, we highlight two representative failure modes that were substantially reduced. First, a common pre-fine-tuning error occurred when LLM provided tool with incorrectly generated tool arguments. The code executor tool requires three explicit arguments including code, input file and output file. However, LLMs frequently generated correct Python code that opened files but failed to pass the filename into the tool's `file_to_load` argument, causing sandbox execution to fail. After fine-tuning, the frequency of such mismatches decreased remarkably. Second, the baseline models tried to use natural-language column names from the task description without checking the provided `metadata.txt` which led to `KeyErrors`. The first step of the fine-tuned models involved examining the metadata file for references to actual column identifiers which led to the development of reliable executable solutions.

**External Validation and Comparison.** To further assess generalization and compare with state-of-the-art specialized agents, we adapt data modeling tasks from DSBench (Jing et al., 2024) into the DARE-bench task format. As shown in Table 8, all fine-tuned versions outperform the original baseline, proving that DARE-bench enhances performance beyond in-domain tasks. Specifically, inclusive sampling methods (AV and DV) yield the most significant improvements by leveraging a wider range of valid traces compared to stricter filtering (FV and BV). Furthermore, we compare our fine-tuned models with DataWiseAgent (You et al., 2025) under identical settings. As detailed in Table 10, our model compare favorably to DataWiseAgent, achieving a score of 42.41 compared to 29.17. This demonstrates that our framework offers competitive adaptability and robustness in diverse data science workflows compared to existing specialized agents.

## 6 CONCLUSION AND FUTURE WORKS

We present DARE-bench, a training-focused benchmark for DS agents which enables executable evaluation and trainable supervision through two verifiable task families: (i) process-aware instruction following with reference-code ground truths, and (ii) ML modeling with dataset ground truths. The 6,300 Kaggle-derived tasks show poor performance from strong general-purpose LLMs until they receive task-specific data but fine-tuning on DARE-bench artifacts produce reliable and repeatable enhancements in process fidelity and predictive performance and execution failure reduction. Our design uses the executable-benchmark approach which software engineering professionals have adopted to solve DS-specific problems that recent evaluations have identified.

We will expand our task type coverage (figures/speeches/clustering), strengthen procedural constraints and verifier-based objectives, and add anomaly detection tracks (tabular and time-series) with appropriate event/segment-level metrics and weak/unsupervised scoring protocols.

---

[4]`https://qwen.readthedocs.io/en/latest/framework/function_call.html`

ACKNOWLEDGMENTS

This work is partially supported by NSF CAREER-2305491. We would like to thank Jeff Rasley for his help with the open-source release. We would like to thank the Area Chair and reviewers for their valuable feedback and suggestions.

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

OVERVIEW OF THE APPENDIX

The Appendix is organized as follows:

- Appendix A contains reproducibility statement.
- Appendix B contains the use of LLMs in this work.
- Appendix C provides the limitation of the work.
- Appendix D contains the explanation of the evaluation metrics used in this work.
- Appendix E provides the performance on the strictly open-sourceable subset.
- Appendix F provides average number of tokens and tool calls for completions of different models and prompts.
- Appendix G provides training details of the RL experiments in this work.
- Appendix H contains detailed description of DARE-bench features.
- Appendix I provides example prompt of the preprocessing steps of this work, including column inference and task identification.
- Appendix J contains reference code for instruction-following tasks in this work.
- Appendix K contains tool schema used in our experiments and some task examples.
- Appendix L provides details of the rejection sampling implementation of this work.
- Appendix M provides details on how we make the use of LLM to classify tasks.

## A    REPRODUCIBILITY STATEMENT

We have attached the subet of our test set in the supplementary materials. Once accepted, we will release the full test set of our benchmark. The training set and model checkpoints will also be provided upon request, and we plan to release them publicly depending on the feedback we receive from the research community. Also, a detailed description of our data processing procedure is included in subsection 3.1. These resources are intended to facilitate reproducibility and allow future researchers to build upon our work.

## B    THE USE OF LARGE LANGUAGE MODELS (LLMS)

In this project, LLMs were used as assistive tools. Specifically, we used LLMs to polish the writing of the paper and to assist in finding related works. In addition, LLMs were used during the data processing stage, for tasks such as data filtering, question rewriting, and identifying task targets. Beyond these uses, the research ideas, experimental design, and analyses were developed independently by the authors. The authors take full responsibility for all content presented in this paper.

## C    LIMITATIONS

While DARE-bench provides a large-scale, verifiable, and trainable benchmark, several limitations remain. First, the current tasks are primarily tabular based, so the benchmark does not yet cover multimodal inputs such as text–image combinations or code–diagram interactions. Second, the cost of generating large numbers of executable traces can be high, and the rejection sampling strategies, while effective, may introduce biases toward shorter trajectories.

## D    EVALUATION METRICS

We report results using two standard metrics for classification and regression tasks: macro-F1 and $R^2$.

**Macro-F1.** For a classification task with $C$ classes, let $\mathrm{TP}_c$, $\mathrm{FP}_c$, and $\mathrm{FN}_c$ denote the number of true positives, false positives, and false negatives for class $c$, respectively. The precision and recall for class $c$ are defined as

$$\mathrm{Precision}_c = \frac{\mathrm{TP}_c}{\mathrm{TP}_c + \mathrm{FP}_c}, \quad \mathrm{Recall}_c = \frac{\mathrm{TP}_c}{\mathrm{TP}_c + \mathrm{FN}_c}.$$

The F1-score for class $c$ is

$$\mathrm{F1}_c = \frac{2 \cdot \mathrm{Precision}_c \cdot \mathrm{Recall}_c}{\mathrm{Precision}_c + \mathrm{Recall}_c}.$$

The macro-F1 is then the unweighted mean across all classes:

$$\mathrm{Macro\text{-}F1} = \frac{1}{C} \sum_{c=1}^{C} \mathrm{F1}_c.$$

$R^2$ **(Coefficient of Determination).** For regression/time-series tasks with ground-truth values $\{y_i\}_{i=1}^n$ and predictions $\{\hat{y}_i\}_{i=1}^n$, define the mean of ground-truth values as $\bar{y} = \frac{1}{n} \sum_{i=1}^n y_i$. The $R^2$ metric is

$$R^2 = 1 - \frac{\sum_{i=1}^n (y_i - \hat{y}_i)^2}{\sum_{i=1}^n (y_i - \bar{y})^2}.$$

An $R^2$ value close to 1 indicates strong predictive performance, while values close to 0 or negative indicate weak or worse-than-baseline performance. Since $R^2$ can be negative when the model performs worse than predicting the mean, we adopt a *clipped* $R^2$ defined as

$$R_{\mathrm{clipped}}^2 = \max(R^2, 0),$$

to ensure that regression scores remain in $[0, 1]$ and are comparable to classification metrics.

## E  PERFORMANCE ON THE STRICTLY OPEN-SOURCEABLE SUBSET

To ensure broad applicability and adherence to strict compliance standards, we constructed a strictly open-sourceable subset of our benchmark. While the full benchmark aggregates data from diverse sources to maximize coverage, certain sources impose licensing constraints (e.g., ShareAlike or Non-Commercial clauses) or lack explicit licensing information, which may limit their utility in proprietary model development.

The strictly open-sourceable subset explicitly excludes data sources governed by restrictive licenses, including *Creative Commons ShareAlike (SA)*, *Non-Commercial (NC)*, and sources with *Unknown* or custom restrictive terms. This subset is composed exclusively of data distributed under permissive licenses, such as *MIT*, *Apache-2.0*, *CC0*, and *CC-BY-4.0*. This ensures that the subset can be freely used, modified, and integrated into downstream applications without "viral" licensing obligations.

Table 11: Performance on the Strictly Open-Sourceable Subset

| Model | class-IF | class-MM | reg-IF | reg-MM | time-XF | time-CF |
|---|---|---|---|---|---|---|
| gpt-4o | 31.86 | 41.10 | 20.63 | 39.74 | 37.09 | 5.24 |
| gpt-4.1 | 55.39 | 57.75 | 50.00 | 57.68 | 41.19 | 6.81 |
| gpt-5 | **70.10** | 43.18 | **55.56** | 55.21 | 36.78 | 7.84 |
| gpt-o4-mini | 68.14 | 59.14 | 51.59 | 57.48 | 42.15 | 8.15 |
| Claude-Sonnet-3.7 | 61.52 | **61.22** | 46.03 | **61.36** | **51.21** | **12.08** |
| Claude-Sonnet-4 | 14.71 | 17.70 | 14.29 | 10.50 | 5.27 | 0.02 |
| Qwen3-32B | 16.67 | 30.92 | 15.08 | 35.42 | 27.26 | 0.00 |
| Qwen3-4B | 3.43 | 4.99 | 0.79 | 2.28 | 7.00 | 0.00 |

Table 11 presents the evaluation results on this filtered subset using the same experimental configuration as the main evaluation (5 turns, 200 s sandbox time).

We observe that the performance trends on this subset are largely consistent with the full benchmark, indicating that the permissive subset retains sufficient difficulty and representativeness to serve as a reliable proxy for the full evaluation.

# F AVERAGE NUMBER OF TOKENS AND TOOL CALLS

Detailed statistics on the average token counts and tool invocations are provided in Table 12. All token metrics are standardized using the Qwen3 tokenizer.

Table 12: Average number of tokens and tool calls for completions of different models and prompts. All token counts are calculated using the Qwen3 tokenizer.

| Model | IF Tokens | IF Tool Calls | MM Tokens | MM Tool Calls | Overall Tokens | Overall Tool Calls |
|---|---|---|---|---|---|---|
| prompt | 596.7 | - | 224.6 | - | 350.7 | - |
| gpt-5 | 609.5 | 2.2 | 582.5 | 2.4 | 591.2 | 2.4 |
| Claude-Sonnet-3.7 | 675.2 | 3.6 | 894.3 | 4.8 | 830.0 | 4.4 |
| Qwen3-32B | 2093.3 | 3.1 | 1693.0 | 3.5 | 1816.8 | 3.4 |
| Qwen3-32B-SFT-DV | 1778.3 | 3.3 | 1572.0 | 3.6 | 1638.7 | 3.5 |
| Qwen3-4B | 1691.1 | 3.7 | 1151.7 | 3.9 | 1328.1 | 3.8 |
| Qwen3-4B-RL | 1549.4 | 3.7 | 1140.1 | 3.7 | 1277.9 | 3.7 |

# G REINFORCEMENT LEARNING TRAINING DETAILS

## G.1 REWARD DESIGN

**Instruction following tasks.** For instruction following tasks including Classification-IF and Regression-IF tasks. We have reference solution code $\mathcal{C}_{\text{ref}}$ with corresponding simulated prediction for data $\mathcal{D}_{\text{test}}$ as $\mathbf{y}_{\text{ref}} = \mathcal{C}_{\text{ref}}(\mathcal{D}_{\text{test}})$. Given the model prediction $\hat{\mathbf{y}} = \mathcal{G}(Q, \mathcal{D}_{\text{train}}, \mathcal{D}_{\text{test}}, M, \mathcal{T})$ and simulated ground truth $\mathbf{y}_{\text{ref}}$, we use the following reward:

$$r = \begin{cases} 0.1, & \hat{\mathbf{y}} \text{ exists}, \\ 1.1, & \hat{\mathbf{y}} = \mathbf{y}_{\text{ref}}, \\ 0, & \text{otherwise}. \end{cases} \tag{1}$$

Note that LLMs may be unable to generate a `prediction.csv` file due to the max turns or sandbox execution time limit.

**Predictive ML tasks.** For other tasks, including classification-PM, regression-PM, time-series-XF, and time-series-CF, we have masked ground-truth data $\mathbf{y}_{\text{gt}}$. Given the prediction provided by LLM $\hat{\mathbf{y}}$, we define the reward as

$$r = \begin{cases} 0.1 + d(\hat{\mathbf{y}}, \mathbf{y}_{\text{gt}}), & \hat{\mathbf{y}} \text{ exists}, \\ 0, & \text{otherwise}, \end{cases} \tag{2}$$

where $d : \mathcal{X} \times \mathcal{Y} \to [0, 1]$ denotes the distance measure between the prediction and the target. For classification tasks, we adopt the macro-F1 score to account for class imbalance. For regression and time-series tasks, we use the *clipped coefficient of determination*, defined as

$$\text{clip}(R^2) = \min\{1, \max\{0, R^2\}\}.$$

If there are multiple prediction targets, we compute the distance by taking the average of them.

## G.2 OTHER TRAINING PARAMETERS

**Reinforcement learning.** We summarize our RL training hyper-parameters in Table 13.

| Hyper-parameter | Value |
|---|---|
| RL algorithm | GRPO (Shao et al., 2024) |
| Loss aggregation | Sequence level |
| Group normalization | False |
| Learning rate | $1 \times 10^{-6}$ |
| Training mini-batch size | 64 |
| KL regularization | False |
| Rollout batch size | 64 |
| Number of rollouts per question | 8 |
| Rollout backend | SGLang (Zheng et al., 2024) |
| Rollout temperature | 1.0 |
| top_p | 0.95 |
| top_k | 50 |
| Model sequence length | 32,768 |

Table 13: Hyper-parameters used for reinforcement learning experiments.

## H  DETAILED DESCRIPTION OF OTHER DARE-BENCH FEATURES

**Automated and Scalable Curation.**   The task generation process in DARE-bench uses a defined approach which collects data from Kaggle and incorporates web-scraped content before LLMs verify the tasks and produce standardized definitions. The automated pipeline generates authentic work assignments at large scale across multiple fields through an approach that needs minimal human involvement.

**Diverse and Realistic Coverage.**   The benchmark contains 6,300 tasks which cover multiple domains and languages, including tabular classification and regression as well as advanced time-series forecasting. By drawing directly from real-world Kaggle datasets, it naturally incorporates common data challenges such as class imbalance, missing values, noise, and temporal irregularities, providing a more faithful simulation of practical data science scenarios.

**Time and interaction constraints.**   DARE-bench implements realistic usage scenarios through its requirement for both time-limited wall-clock operation and restricted interaction turn counts. In practice, end users are unlikely to wait hours for a model to train a full pipeline; hence, we cap execution time to 10 minutes for fast-response settings. The system limits the total number of agent-environment dialogues which forces models to find efficient solutions instead of performing endless exploration. The established limitations in this benchmark create a testing environment which mirrors actual operational conditions for interactive data science agents.

# I EXAMPLE PROMPT FOR COLUMN INFERENCE AND TASK IDENTIFICATION

The following prompt guides the model to check task suitability and identify prediction target and relevant features from the provided dataset description and data information.

---

**Task Target and Feature Identification Prompt**

You are given a dataset along with its description.
Your tasks are as follows:

**Task 1: Assess Logistic/Linear Regression Suitability**
Determine whether logistic regression (for classification) or linear regression (for regression) can be appropriately used to model this dataset.
Use this strict checklist:

- Classification: target must be categorical; features structured; manageable missing values.

- Regression: target must be numeric and continuous; features structured; manageable missing values; categorical targets not allowed.

If all conditions are met, the method is appropriate. Otherwise, it is not. When uncertain, output False.

**Task 2: Identify Task Type, Target Column, and Feature Columns**
You must select column names only from the list below inside *Column list:*, avoid using names from *Context / description:*.

```
Column list:
{all_columns}

Context / description:
{filtered_metadata}
{scraper}
```

Infer:

- The task type (classification or regression)

- A list ($\leq 3$) of candidate target columns

- The best set of feature columns

**Task 3: Column Type Inference**
For each column in the list, classify it as:

- "numerical": meaningful arithmetic operations

- "categorical": groups/codes, arithmetic not meaningful

Instructions:

- Return a Python dictionary with every column as a key

- Value must be either "numerical" or "categorical"

- Use dataset description to guide decisions

**Final Output Format**
Output exactly 5 lines, in LaTeX-boxed format:
1. Method suitability → `\boxed{True}` or `\boxed{False}`
2. Task type → `\boxed{classification}` or `\boxed{regression}`
3. Target column candidates → `\boxed{["target1", "target2"]}`
4. Feature columns → `\boxed{["col1", "col2", ...]}`
5. Column types → `\boxed{{{"col1": "numerical", "col2": "categorical"}}}`

---

The following prompt reformulates the user question into a precise and well-structured instruction.

---

**Rewrite Question Prompt**

You are given a machine learning task described in `final_question` and a dictionary of column metadata in `metadata_description`. Your job is to rewrite the `final_question` in fluent natural language, making it easier to read while keeping the meaning and structure intact.
Here's what you must do:

1. Replace all column names and feature names in `final_question` with their natural language descriptions from `metadata_description`. Preserve the original ordering of features in lists.

2. If a column or feature name is not present in `metadata_description`, rewrite it into a natural-sounding phrase using best judgment.

3. Rewrite structured formats (lists, dicts) into natural language paragraphs, while retaining original item order.

4. Keep existing natural language unchanged.

5. Keep all file paths unchanged.

6. File names or paths must be wrapped in backticks.

7. Target column names must be wrapped in backticks.

8. Final output must be a clear instruction in natural language.

9. If the string `None` appears in value ranges, treat it as a categorical value `None`.

10. Do not include headings, markdown, or extra explanations—return only the rewritten question.

11. Use only standard English characters.

12. Explicitly preserve ordering requirements in the rewritten question.

- - -

Here is the `final_question`:

{question}

- - -

Here is the `metadata_description`:

{description}

- - -

Now return only the rewritten version of the question, using natural language descriptions where possible. Preserve file paths, model names, and categorical values exactly as given.

The following prompt determines whether the dataset is time-series and infers the appropriate temporal type information.

---

**Time-Series Identification and Typing Prompt**

You are given a dataset and its description.

**Task 1: Assess Suitability**
Check if Time Series Forecasting applies. Conditions:

1. Must have a clear timestamp column (e.g., 'date', 'time').

2. Must have a target variable changing over time (e.g., sales, temperature).

3. Observations should be sequential and time-dependent.

4. Time interval must be regular or resample-able (e.g., daily, hourly).

If all are met, output True; otherwise False.

**Task 2: Identify Key Columns**
From the list below, infer:

- Best **timestamp column**

- Best **target column**

- Optional exogenous feature columns

```
Column list:
{all_columns}

Context:
{filtered_metadata}
{scraper}

Preview (first 50 rows):
{df_preview}
```

**Task 3: Column Typing**
For each column, classify as: `"timestamp"`, `"numerical"`, `"categorical"`, or `"other"`.
If timestamp exists, also infer its format (Python strftime).

**Final Output Format** (6 lines, LaTeX-boxed):

1. Suitability → \boxed{True} or \boxed{False}

2. Timestamp column → \boxed{column_name} or \boxed{ambiguous}

3. Target column → \boxed{column_name} or \boxed{ambiguous}

4. Exogenous features → \boxed{["col1", ...]} or \boxed{[]}

5. Column types → \boxed{{"col1": "timestamp", "col2": "numerical"}}

6. Time format → \boxed{%Y-%m-%d %H:%M:%S} or \boxed{ambiguous}

---

The following prompt identifies grouping entities (e.g., users, products, or regions) that structure the dataset for time-CF tasks.

---

**Entity (Group) Identification Prompt**

You are given a dataset for time series forecasting. The dataset includes a timestamp column, a target column to be predicted, and possibly multiple other columns representing categorical or numeric features. Your task is to identify which column(s) represent the entity (or group ID) — that is, the column(s) that differentiate multiple independent time series within the dataset.

Please analyze the column names and a sample of the data (including at least the first few rows), and answer the following:

1. Which column(s) should be used to distinguish different time series entities?

2. Briefly explain why those column(s) were selected as entity identifiers.

3. If no entity column is needed because the dataset represents a single time series, say so explicitly.

---

**Dataset Description**

`{description}`

---

**Additional identification suggestions (optional)**

`{entity_identification_suggestions}`

---

**Sample Data (first 30 rows)**

`{sample_str}`

---

**Column statistics (distinct counts, top value frequency, example value patterns)**

`{column_stats_str}`

---

**Output format (exactly 2 lines, LaTeX-boxed, nothing else):**

1. Entity Columns → `\boxed{["col_name1", "col_name2", ...]}` or `\boxed{[]}` if none

2. Justification → `\boxed{<Your explanation here>}`

The following prompt decides whether resampling is needed for the dataset and, if so, specifies the appropriate resampling strategy.

---

**Resampling Decision Prompt**

You are an expert data scientist assisting with time series preprocessing.
Your goal is to decide whether a given time series dataset needs **resampling** (i.e., converting irregular or overly fine-grained timestamps to a fixed frequency like daily/hourly).

**Task Description**
You are given:

1. A **brief description** of the dataset and task.

2. The **first 30 rows** of the dataset (including timestamps and relevant columns).
   - For each time value, up to 5 rows are shown.
   - If a time value had more than 5 rows, it is marked with a comment.

3. The **target column** for forecasting or analysis: {target_col}.

Please analyze:

- Whether the time column appears **irregular**, **too granular**, or **dense**.
- Whether each row represents a **meaningful unit** (e.g., per-day summary) or a **low-level log** (e.g., events).
- Whether **resampling** could make the series easier to model.
- If resampling is needed, recommend:
  - The **resampling rule** (e.g., `1min`, `5min`, `1H`, `1D`).
  - The **aggregation function** for the target column ({target_col}): choose from `"mean"`, `"sum"`, `"count"`, or other common aggregations.
- If resampling is not needed, it may be because the data is evenly spaced or each row is meaningful as-is.

---

**Dataset Description**

`{description}`

---

**Sample Data (first 30 rows)**

`{sample_str}`

---

**Output format (exactly 4 lines, LaTeX-boxed, nothing else):**

1. Should resample → \boxed{True} or \boxed{False}
2. Reason → \boxed{<One-sentence explanation>}
3. Suggested rule → \boxed{<1min/5min/1H/1D or null>}
4. Target aggregation → \boxed{<mean/sum/count/... for '{target_col}'>}

## J  REFERENCE CODE FOR INSTRUCTION-FOLLOWING EVALUATION

Below we include the reference implementation used to evaluate instruction-following tasks in our benchmark.

---

**Reference Code for Instruction-Following Evaluation**

```python
import json
import os
import sqlite3
import pandas as pd
import numpy as np
from sklearn.model_selection import train_test_split
from sklearn.preprocessing import StandardScaler, OneHotEncoder
from sklearn.impute import SimpleImputer
from sklearn.compose import ColumnTransformer
from sklearn.pipeline import Pipeline
from sklearn.linear_model import LogisticRegression, LinearRegression, Ridge, Lasso
from sklearn.tree import DecisionTreeClassifier, DecisionTreeRegressor
from sklearn.neighbors import KNeighborsClassifier, KNeighborsRegressor
from sklearn.neural_network import MLPClassifier, MLPRegressor
from sklearn.naive_bayes import GaussianNB
from sklearn.svm import LinearSVC
from sklearn.multioutput import MultiOutputClassifier
from sklearn.metrics import accuracy_score, classification_report, mean_squared_error,
     r2_score

# Function to load and join tables from a SQLite file
def load_and_join(sqlite_path):
    conn = sqlite3.connect(sqlite_path)
    tables = pd.read_sql_query("SELECT name FROM sqlite_master WHERE type='table';",
         conn)['name'].tolist()
    df = None
    for table in tables:
        df_tab = pd.read_sql_query(f"SELECT * FROM '{table}'", conn)
        if 'row_id' not in df_tab.columns:
            continue
        if df is None:
            df = df_tab
        else:
            df = df.merge(df_tab, on='row_id', how='inner')
    conn.close()
    return df

def train_predict_model(train_df, eval_df, feature_cols, model_type,
     column_type_inference, target_cols=['answer'], imputer_type="most_frequent",
                        problem_type="classification", random_state=42):
    print(f"MACHINE LEARNING PIPELINE")
    print("=" * 60)

    # Check if target column exists in training data
    for target_column in target_cols:
        if target_column not in train_df.columns:
            print(f"Target column '{target_column}' not found in training data!")
            print(f"Available columns in train_df: {list(train_df.columns)}")
            return None

    # Check if all feature columns exist in both datasets
    missing_features_train = [col for col in feature_cols if col not in train_df.
         columns]
    missing_features_eval = [col for col in feature_cols if col not in eval_df.columns]

    if missing_features_train:
        print(f"Missing feature columns in train_df: {missing_features_train}")
        return None

    if missing_features_eval:
        print(f"Missing feature columns in eval_df: {missing_features_eval}")
        return None

    formatted_targets = ", ".join("'{}'".format(col) for col in target_cols)
    print(f"Target column {formatted_targets} found in training data")
    print(f" Training dataset shape: {train_df.shape}")
    print(f" Evaluation dataset shape: {eval_df.shape}")

    # Prepare training features and target
```

```
    X_train = train_df[feature_cols].copy()
    y_train = train_df[target_cols].copy()

    # Prepare evaluation features (and target if it exists)
    X_eval = eval_df[feature_cols].copy()

    # Check if target column exists in eval_df for evaluation
    has_eval_target = True
    for target_column in target_cols:
        if target_column not in eval_df.columns:
            has_eval_target = False
    if has_eval_target:
        y_eval = eval_df[target_cols].copy()
        print(f"Target column found in evaluation data - will compute metrics")
    else:
        y_eval = None
        print(f" Target column not found in evaluation data - will only make
            predictions")

    # Check for null targets in training data
    null_targets_train = y_train.isnull().sum().sum()  # use two sum to get the total
         null number
    if null_targets_train > 0:
        print(f" Found {null_targets_train} null targets in training data - removing
            these rows")
        valid_indices = ~y_train.isnull().any(axis=1)  # make sure no null target row
        X_train = X_train[valid_indices]
        y_train = y_train[valid_indices]

    print(f" Final training data: {X_train.shape[0]} rows")
    print(f" Final evaluation data: {X_eval.shape[0]} rows")

    # Separate numeric and categorical features
    numeric_features = []
    categorical_features = []

    for col in feature_cols:
        # Check data type in training data
        if column_type_inference[col].lower() == "numerical":
            numeric_features.append(col)
        elif column_type_inference[col].lower() == "categorical":
            categorical_features.append(col)

    print(f" Numeric features ({len(numeric_features)}): {numeric_features}")
    print(f" Categorical features ({len(categorical_features)}): {categorical_features
        }")
    assert len(numeric_features) + len(categorical_features) == len(feature_cols)

    # Create preprocessing pipeline
    transformers = []

    if numeric_features:
        transformers.append(('num', Pipeline([
            ('imputer', SimpleImputer(strategy=imputer_type)),
            ('scaler', StandardScaler())
        ]), numeric_features))

    if categorical_features:
        transformers.append(('cat', Pipeline([
            ('imputer', SimpleImputer(strategy="most_frequent")),
            ('onehot', OneHotEncoder(handle_unknown='ignore', sparse_output=False))
        ]), categorical_features))

    preprocessor = ColumnTransformer(transformers=transformers)

    # Choose model based on problem type
    if problem_type.lower() == "classification":
        if model_type == "LogisticRegression":
            model = LogisticRegression(random_state=random_state)
            if len(target_cols) > 1:
                model = MultiOutputClassifier(model)
        elif model_type == "DecisionTreeClassifier":
            model = DecisionTreeClassifier(random_state=random_state)
        elif model_type == "GaussianNB":
            model = GaussianNB()
            if len(target_cols) > 1:
                model = MultiOutputClassifier(model)
        elif model_type == "LinearSVC":
```

```
            model = LinearSVC(random_state=random_state)
            if len(target_cols) > 1:
                model = MultiOutputClassifier(model)
        elif model_type == "MLPClassifier":
            model = MLPClassifier(random_state=random_state)
        else:
            raise ValueError(f"Invalid model type: {model_type}")
        print(f" Using {model_type} for classification")

    elif problem_type.lower() == "regression":
        if model_type == "LinearRegression":
            model = LinearRegression()
        elif model_type == "DecisionTreeRegressor":
            model = DecisionTreeRegressor(random_state=random_state)
        elif model_type == "Ridge":
            model = Ridge(random_state=random_state)
        elif model_type == "Lasso":
            model = Lasso(random_state=random_state)
        elif model_type == "MLPRegressor":
            model = MLPRegressor(random_state=random_state)
        else:
            raise ValueError(f"Invalid model type: {model_type}")

        print(f" Using {model_type} for regression")

    # Create full pipeline
    ml_pipeline = Pipeline([
        ('preprocessor', preprocessor),
        ('model', model)
    ])

    # Train the model
    print(f"TRAINING MODEL...")
    try:
        ml_pipeline.fit(X_train, y_train)
        print(f" Model trained successfully")
    except Exception as e:
        print(f" Error during training: {e}")
        return None

    # Make predictions
    print(f"MAKING PREDICTIONS...")
    try:
        y_pred = ml_pipeline.predict(X_eval)
        print(f"Predictions completed")
    except Exception as e:
        print(f"Error during prediction: {e}")
        return None

    # Evaluate model (only if we have evaluation targets)
    if has_eval_target and y_eval is not None:
        print(f"MODEL EVALUATION")
        print("=" * 30)

        # Remove rows with null targets in evaluation for metrics
        valid_eval_mask = y_eval.notna()
        y_eval_clean = y_eval[valid_eval_mask]
        y_pred_clean = y_pred[valid_eval_mask]

        if len(y_eval_clean) == 0:
            print(" No valid evaluation targets found - skipping evaluation metrics")
        else:
            if problem_type.lower() == "classification":
                accuracy = accuracy_score(y_eval_clean, y_pred_clean)
                print(f" Accuracy: {accuracy:.4f}")
                print(f"Classification Report:")
                print(classification_report(y_eval_clean, y_pred_clean))

                # Show sample predictions
                print(f"Sample Predictions:")
                for i in range(min(10, len(y_eval_clean))):
                    actual = y_eval_clean.iloc[i]
                    predicted = y_pred_clean[i]
                    status = "right" if actual == predicted else "wrong"
                    print(f"{status} Row {i}: Actual={actual}, Predicted={predicted}")

            else:
```

```
                    mse = mean_squared_error(y_eval_clean, y_pred_clean)
                    r2 = r2_score(y_eval_clean, y_pred_clean)
                    rmse = np.sqrt(mse)

                    print(f" R^{2} Score: {r2:.4f}")
                    print(f" RMSE: {rmse:.4f}")
                    print(f" MSE: {mse:.4f}")

                    # Show sample predictions
                    print(f"Sample Predictions:")
                    for i in range(min(10, len(y_eval_clean))):
                        actual = y_eval_clean.iloc[i]
                        predicted = y_pred_clean[i]
                        diff = abs(actual - predicted)
                        print(f"   Row {i}: Actual={actual:.3f}, Predicted={predicted:.3f},
                              Diff={diff:.3f}")
        else:
            print(f"EVALUATION SKIPPED - No target column in evaluation data")
            print(f" Generated {len(y_pred)} predictions")

        y_pred_df = pd.DataFrame(y_pred, columns=y_train.columns)
        y_pred_df.insert(0, 'row_id', eval_df["row_id"].values)
        return {
            'pipeline': ml_pipeline,
            'predictions': y_pred_df,
            'eval_indices': X_eval.index,
            'problem_type': problem_type,
            'X_train': X_train,
            'y_train': y_train,
            'X_eval': X_eval,
            'y_eval': y_eval if has_eval_target else None,
            'has_eval_target': has_eval_target,
            "numeric_features": numeric_features,
            "categorical_features": categorical_features,
        }

feature_cols = $feature_cols
model_type = "$model_type"
column_type_inference = $column_type_inference
target_cols = $target_cols
imputer_type = "$imputer_type"
problem_type = "$problem_type"
random_state = $random_state
save_file_type = "$save_file_type"

if save_file_type == 'sqlite':
    conn = sqlite3.connect("train_v1_no_err.sqlite")
    train_df = pd.read_sql("SELECT * FROM train_set", conn)
    train_df = train_df.replace({None: np.nan})
    conn.close()

    eval_df = load_and_join("val_v1.sqlite")
    eval_df = eval_df.replace({None: np.nan})
elif save_file_type == 'csv':
    train_df = pd.read_csv('train_v1_no_err.csv', keep_default_na=False, na_values
        =[""])
    eval_df = pd.read_csv('val_v1.csv', keep_default_na=False, na_values=[""])
elif save_file_type == 'parquet':
    train_df = pd.read_parquet('train_v1_no_err.parquet')
    train_df = train_df.replace({None: np.nan})
    eval_df = pd.read_parquet('val_v1.parquet')
    eval_df = eval_df.replace({None: np.nan})

result = train_predict_model(
    train_df = train_df,
    eval_df = eval_df,
    feature_cols = feature_cols,
    model_type = model_type,
    column_type_inference=column_type_inference,
    target_cols=target_cols,
    imputer_type=imputer_type,
    problem_type=problem_type,
    random_state=random_state
)

result['predictions'].to_csv('simulated_pred_local.csv', index=False)
```

# K  TOOL SCHEMA AND TASK EXAMPLES

The following schema defines the details of our code executor tool.

```
Tool Schema: python_executor

{
  "type": "function",
  "name": "python_executor",
  "description": "Execute a Python script in an isolated HTTP sandbox with a 200-second
      time limit. Each run is single-shot and stateless (no REPL, no persistent
      environment between runs). You may upload input files via `files_to_load` and
      retrieve results via `files_to_save`. The maximum file size for both upload and
      download is 200 MB. The tool returns the full program output, including both
      stdout and stderr. Use explicit `print(...)` statements to capture values in the
       output. This tool can be invoked up to 3 times per conversation.",
  "parameters": {
    "type": "object",
    "properties": {
      "code": {
        "type": "string",
        "description": "The Python code to execute."
      },
      "files_to_load": {
        "type": "array",
        "items": {
          "type": "string"
        },
        "description": "List of input file paths to upload prior to execution (e.g. [\"
            input1.csv\", \"config.json\"])."
      },
      "files_to_save": {
        "type": "array",
        "items": {
          "type": "string"
        },
        "description": "List of output file paths to download after execution (e.g. [\"
            results.csv\", \"log.txt\"])."
      }
    },
    "required": [
      "code",
      "files_to_load",
      "files_to_save"
    ],
    "additionalProperties": false
  },
  "strict": true
}
```

The following provides a example of IF task.

---

**Instruction Following Task Example**

**Question**

Please complete the task as described below without asking any follow-up questions or requesting additional information. Proceed under the assumption that all required information is provided. You are given access to a training Parquet file named `train_v1.parquet`, which contains a single table, and a metadata file `metadata.txt` that describes the original dataset and each of its columns. Your task is to perform classification using this data and predict the target column `Personality` for the validation set located at `val_v1.parquet`. Unless stated otherwise, you should use default parameters for all steps including model training and preprocessing. First, load the training file directly. Then, filter the training dataset using the expected ranges while retaining any rows that have missing values in the relevant columns, excluding only those rows where a non-missing value violates its expected range. The expected ranges are as follows in the specified order: number of close friends must be between 0.0 and 15.0; presence of stage fright must be either "No" or "Yes"; social media post frequency must be between 0.0 and 10.0; frequency of going outside must be between 0.0 and 7.0; feeling drained after socializing must be either "No" or "Yes"; frequency of social events must be between 0.0 and 10.0; and hours spent alone daily must be between 0.0 and 11.0. After filtering, select only the features listed in their original order: number of close friends, frequency of social events, presence of stage fright, feeling drained after socializing, and hours spent alone daily. The numeric features, to be used in the specified order, are number of close friends, frequency of social events, and hours spent alone daily, and the categorical features, also to be used in the specified order, are presence of stage fright and feeling drained after socializing. Handle missing values by imputing numeric features with the mean and categorical features with the most frequent value. Preprocess the data by applying a standard scaler to the numeric features and one-hot encoding to the categorical features with `handle_unknown` set to `ignore` and `sparse_output` set to `False`. Train a single `LogisticRegression` model using scikit-learn with `random_state=86`. Finally, make predictions on the validation set and save the results to a CSV file at `prediction.csv`, including the column `row_id` as provided in the original `val_v1.parquet` and the corresponding predictions aligned with each `row_id` so that performance can be computed correctly.

**Metadata**

Overview

*Dive into the Extrovert vs. Introvert Personality Traits Dataset, a rich collection of behavioral and social data designed to explore the spectrum of human personality. This dataset captures key indicators of extroversion and introversion, making it a valuable resource for psychologists, data scientists, and researchers studying social behavior, personality prediction, or data preprocessing techniques.*

Context

*Personality traits like extroversion and introversion shape how individuals interact with their social environments. This dataset provides insights into behaviors such as time spent alone, social event attendance, and social media engagement, enabling applications in psychology, sociology, marketing, and machine learning. Whether you're predicting personality types or analyzing social patterns, this dataset is your gateway to uncovering fascinating insights.*

Dataset Details
**Size**: The dataset contains 2,900 rows and 8 columns.
**Features**:

- `Time_spent_Alone`: Hours spent alone daily (0–11).
- `Stage_fear`: Presence of stage fright (Yes/No).
- `Social_event_attendance`: Frequency of social events (0–10).
- `Going_outside`: Frequency of going outside (0–7).
- `Drained_after_socializing`: Feeling drained after socializing (Yes/No).
- `Friends_circle_size`: Number of close friends (0–15).
- `Post_frequency`: Social media post frequency (0–10).
- `Personality`: Target variable (Extrovert/Introvert).

Data Quality: Includes some missing values, ideal for practicing imputation and preprocessing.

Format: Single CSV file, compatible with Python, R, and other tools.

Data Quality Notes

---

- Contains missing values in columns like `Time_spent_Alone` and `Going_outside`, offering opportunities for data cleaning practice.
- Balanced classes ensure robust model training.
- Binary categorical variables simplify encoding tasks.

Potential Use Cases

- Build machine learning models to predict personality types.
- Analyze correlations between social behaviors and personality traits.
- Explore social media engagement patterns.
- Practice data preprocessing techniques like imputation and encoding.
- Create visualizations to uncover behavioral trends.

=== About this file ===

About this file This dataset contains 2,900 entries with 8 features related to social behavior and personality traits, designed to explore and classify individuals as Extroverts or Introverts.

The following provides an example of ML Modeling task.

---

### ML Modeling Task Example

**Question**

Please complete the task as described below without asking any follow-up questions or requesting additional information. Your task is to achieve good performance while balancing training time and accuracy in the sandbox. You are provided with a processed dataset, along with a metadata file `metadata.txt` that describes the original dataset and each of its columns. Load data in `train_v2.parquet` as your full training set. Once that's done, using only CPU resources, train a classification model on this training data. Then use `val_v2.parquet` to generate a prediction for `Internet_Access` for each `row_id`. The model should output a file named `prediction.csv`. The file must contain the column `row_id` (as provided in the original `val_v2.parquet` and the corresponding predictions. Each prediction should be aligned with its row_id so that performance can be computed correctly.

**Metadata**

AI Tool Usage by Indian College Students 2025

This unique dataset, collected via a May 2025 survey, captures how 496 Indian college students use AI tools (e.g., ChatGPT, Gemini, Copilot) in academics. It includes 16 attributes like AI tool usage, trust, impact on grades, and internet access, ideal for education analytics and machine learning.
Columns

- `Student_Name`: Anonymized student name.
- `College_Name`: College attended.
- `Stream`: Academic discipline (e.g., Engineering, Arts).
- `Year_of_Study`: Year of study (1–4).
- `AI_Tools_Used`: Tools used (e.g., ChatGPT, Gemini).
- `Daily_Usage_Hours`: Hours spent daily on AI tools.
- `Use_Cases`: Purposes (e.g., Assignments, Exam Prep).
- `Trust_in_AI_Tools`: Trust level (1–5).
- `Impact_on_Grades`: Grade impact (-3 to +3).
- `Do_Professors_Allow_Use`: Professor approval (Yes/No).
- `Preferred_AI_Tool`: Preferred tool.
- `Awareness_Level`: AI awareness (1–10).
- `Willing_to_Pay_for_Access`: Willingness to pay (Yes/No).
- `State`: Indian state.
- `Device_Used`: Device (e.g., Laptop, Mobile).
- `Internet_Access`: Access quality (Poor/Medium/High).

Use Cases - Predict academic performance using AI tool usage. - Analyze trust in AI across streams or regions. - Cluster students by usage patterns. - Study digital divide via 'Internet_Access'.
Source: Collected via Google Forms survey in May 2025, ensuring diverse representation across India.
Note: First dataset of its kind on Kaggle!

The following provides an example of Time Series Canonical Forecasting Task.

---

**Time Series Canonical Forecasting Task Example**

**Question**

Please complete the task as described below without asking any follow-up questions or requesting additional information. Your task is to achieve good performance while balancing time and accuracy in the sandbox environment. You are provided with a processed dataset, along with a metadata file `metadata.txt` that describes the original dataset and each of its columns. Load the file `train.csv` as your complete training data. This file contains the raw, non-resampled time series data. Once that's done, using only CPU resources, train time series analysis model(s) on the training data. Then, based on the file val_v2.csv, forecast the target column `Close`. Generate predictions exactly for each row_id present in the validation data, Output a file named `prediction.csv`. The file must contain the column row_id (as provided in the original `val_v2.csv` and the corresponding predictions. Each prediction should be aligned with its row_id so that performance can be computed correctly.

**Metadata**

Here's to the crazy ones—the data dreamers, the analysts, the visionaries who believe that a handful of numbers can reveal the DNA of innovation. This dataset is more than a collection of Apple Inc.'s historical stock prices; it's a chronicle of invention, perseverance, and thinking differently.

What's Inside
- Time Span: Daily stock price data for Apple Inc. over multiple years - Features: - 'Date': The day of the record - 'Close': Price at market close - Format: CSV, clean and ready for analysis

Why This Matters
Apple is not just a company, it's a movement. Its stock price reflects not only financial performance, but the world's response to innovation—launches, leadership changes, economic cycles, and the occasional "one more thing."

Possibilities
- Visualize long-term growth and volatility - Model trends, moving averages, or momentum - Forecast future prices with machine learning - Detect the impact of major product launches or events - Explore relationships between volume and price action

Inspiration
As you explore this data, don't just look for patterns—look for stories. See how moments of genius and risk-taking ripple through the numbers. Use this dataset to inspire your own creativity, your own analysis, your own 'insanely great' discoveries.

Whether you're here to build a predictive model, craft beautiful visualizations, or simply marvel at the journey, remember: The people who are crazy enough to think they can change the world with data. . . are the ones who do.

=== About this file ===
About this file This file contains historical daily stock price data for Apple Inc. Each row represents one trading day and includes key financial metrics that track Apple's performance on the stock market.

=== Columns & descriptions ===
Date: The calendar date for the trading record (format: YYYY-MM-DD). Close: The price of Apple's stock at the end of the trading day.

---

## L    REJECTION SAMPLING IMPLEMENTATION DETAILS

We sample up to $K=8$ candidate trajectories per task. Each trajectory records: (i) `final_score` and (ii) end-to-end wall-clock `time`. For IF tasks, `final_score` is exact match $\in \{0,1\}$; for predictive tasks, `final_score` is a normalized metric such as macro-F1 or clipped $R^2$.

### L.1   VALIDITY AND DIVERSITY CONDITIONS

**Validity.**    A trajectory is considered *valid* if:

- For IF tasks: `final_score` $= 1$.
- For predictive tasks: `final_score` $\geq$ type-specific threshold:

$$\text{class-MM: } 0.8, \quad \text{reg-MM: } 0.7, \quad \text{time-XF: } 0.6, \quad \text{time-CF: } 0.3.$$

**Diversity.**    A task is considered *diverse* if:

- For IF tasks: among the $K$ trials, at least one `final_score` $= 1$ and at least one `final_score` $= 0$.
- For predictive tasks: the variance of the $K$ scores satisfies

$$\text{Var}(S_i) \geq \text{threshold}, \quad \text{class-MM/reg-MM: } 0.15, \text{ time-XF: } 0.15, \text{ time-CF: } 0.1.$$

### L.2   REJECTION SAMPLING STRATEGIES

**FV (Fastest-Valid).**    For every task that has at least one valid trajectory:

- IF tasks: keep the single fastest valid trajectory.
- Predictive tasks: keep the trajectory with the highest `final_score`.

**AV (All-Valid).**    For every task:

- Keep all valid trajectories (as defined above).

**BV (Best-Valid).**    For every *diverse* task:

- IF tasks: keep the single fastest valid trajectory.
- Predictive tasks: keep the trajectory with the highest `final_score`.

Thus BV applies the same selection rule as FV, but restricted to diverse tasks only.

**DV (Duo-Valid).**    For every *diverse* task:

- IF tasks: keep the two fastest valid trajectories (or one if fewer exist).
- Predictive tasks: keep the top-2 trajectories by score, restricted to those with $s(t) > \overline{s}_i$ (above mean).

NOTES

- FV applies to *all tasks* with valid traces; BV and DV apply only to *diverse tasks*.
- AV is the only strategy that may return multiple valid trajectories even for non-diverse tasks.
- FV/BV always select at most one trajectory; DV at most two; AV can return more.
- This design ensures a balance between efficiency (FV), diversity (AV), quality (BV), and complementary coverage (DV).

# M    TASK DOMAIN CLASSIFICATION METHODOLOGY

To assess the diversity of DARE-bench and verify its coverage across real-world scenarios, we classified each task into a primary domain (e.g., Finance, Health, Technology). Given the scale of the benchmark (6,300 tasks), manual classification was infeasible. Therefore, we employed an automated LLM-based classification pipeline utilizing the rich metadata associated with each Kaggle-derived dataset.

**Metadata Usage.** The classification relies on four key metadata fields:

- **Title:** The official name of the dataset.
- **Subtitle:** A short phrase summarizing the dataset content.
- **Description:** The full natural language description of the dataset context.
- **Keywords:** User-provided tags from the original Kaggle source.

**Classification Taxonomy.** To ensuring consistency, we defined a controlled vocabulary of allowed domains based on common industry verticals: *finance, health, business, technology, automotive, education, environment*, and *others*.

**Prompt Design.** We constructed a strict prompt to instruct the LLM to identify the single best domain label. The prompt enforces a hierarchical reasoning logic: it prioritizes explicit domain terms found in the user-provided keywords before inferring the domain from the title or description. The full prompt template is provided below:

---

**Domain Classification Prompt**

**Instruction:** Identify the single best domain for a task using the provided metadata.

**Input Data:**
- Title: {title}
- Subtitle: {subtitle}
- Description: {description}
- Keywords: {keywords}

**Example Domains:** [agriculture, finance, health, business, technology, automotive, education, environment, other]

**Reasoning Steps:**
1. **Keyword Check:** First, strictly check the provided 'keywords' list for any explicit domain word (e.g., 'finance', 'health'). If a match is found from the allowed list, select it immediately.
2. **Inference:** If no direct domain appears in the keywords, infer the most appropriate domain based on the semantic context of the 'title', 'subtitle', and 'description'.
3. **Output Formatting:** Output exactly ONE lowercase word from the allowed domains list. Do not output punctuation, explanations, or spaces. If the domain is uncertain or does not fit the specific categories, output 'other'.

**Output:**

---

