# OpenReview forum: "DARE-bench: Evaluating Modeling and Instruction Fidelity of LLMs in Data Science"
_ICLR.cc/2026/Conference — ICLR 2026 Poster_

### Official Review · Reviewer_Ffce · 2025-10-23

**Soundness:** 3
**Presentation:** 3
**Contribution:** 3
**Rating:** 8
**Confidence:** 4

**Summary:**

The author suggests DARE-bench, designed to evaluate and train LLMs for data science workflows. It addresses two major gaps in existing benchmarks: (1) lack of process-aware evaluation capturing instruction adherence and modeling fidelity, and (2) lack of accurately labeled training data. DARE-bench includes 6,300 verifiable, executable Kaggle-derived tasks divided into two families. The benchmark enables reproducible automatic evaluation via sandboxed code execution. Experiments show that general-purpose LLMs perform poorly on these tasks without domain-specific training, but fine-tuning on DARE-bench data dramatically improves both process fidelity and prediction accuracy. The paper demonstrates up to 8× improvement in smaller models after reinforcement learning fine-tuning, and also validates generalization improvements on external datasets like DSBench.

**Strengths:**

1. Introduces a large-scale, executable, and verifiable benchmark that measures both process fidelity and predictive accuracy, filling a key gap in LLM evaluation for data science.

2. Provides a rich dataset derived from Kaggle that supports both evaluation and supervised/RL-based training, improving reproducibility and scalability.

3. Demonstrates concrete, measurable performance improvements and reduced execution failures after fine-tuning, validating benchmark utility for both evaluation and model training.

4. Clearly integrates determinism and sandbox execution to ensure fair, reproducible comparison across models, a major methodological advantage.

**Weaknesses:**

1. Heavy reliance on Kaggle-derived data may bias task diversity toward structured tabular and forecasting problems, limiting broader domain generalization.

2. The benchmark’s focus on reproducibility via deterministic setups may underrepresent realistic data science variability or stochastic modeling behavior.

3. The paper lacks ablation or error analysis to quantify which aspects of fine-tuning drive improvements most.

**Questions:**

1. How does DARE-bench handle tasks involving stochastic algorithms (e.g., random forest, neural nets) while maintaining deterministic evaluation?

2. Are the reference “ground truth” codes verified manually, or automatically synthesized and how is correctness guaranteed?

3. Does reinforcement learning with DARE-bench tasks risk overfitting to procedural templates rather than improving general data reasoning?

4. Can this benchmark be extended beyond Kaggle-style tabular tasks, e.g., to unstructured text or multimodal datasets?

---

> ### Author Response · Authors · 2025-11-23
> **Response to reviewer Ffce [Part 1/3]**
>
> We sincerely appreciate the reviewer for the positive and thoughtful feedback. Below we address each comment in detail.
>
> > heavy reliance on Kaggle-derived data may bias task diversity toward structured tabular and forecasting problems, limiting broader domain generalization.
>
> We thank the reviewer for pointing out this. We respectfully clarify that the use of Kaggle data is a strategic choice, considering reliability and standardization.
>
> 1. **Sourcing from Kaggle aligns with existing standards and ensures rigorous evaluation mechanics.** Following the settings of well-established benchmarks [1,2], we selected Kaggle as our source to ensure scandalization and quality. This choice guarantees that all tasks possess clear licensing, rich metadata, and verifiable ground truths, which are essential for the reproducible, automated evaluation of data science agents. In addition, we categorized the dataset into distinct domains (e.g. Finance, Health, Marketing). As shown in the table below, this demonstrates that our benchmark covers a wide range of real-world scenarios.
>
> Table 1：Domain distribution across DARE-bench
> | Dataset | Finance | Health | Business | Technology | Automotive | Education | Environment | Others |
> | ----- | ----- | ----- | ----- | ----- | ----- | ----- | ----- | ----- |
> | Train set | 16.9% | 10.2% | 7.3% | 4.0% | 4.5% | 2.8% | 6.8% | 47.5% |
> | Test set | 17.1% | 8.4% | 8.2% | 5.6% | 3.3% | 3.1% | 2.4% | 51.9% |
>
> 2. **The DARE-bench curation pipeline is designed to facilitate future expansion to other sources.** Our curation and verification pipeline is not limited to Kaggle. It is an extensible framework capable of processing tasks from a broad range of sources, as long as reliable data and verifiable ground truth are provided. We view our current dataset as a foundation and are planning to leverage this extensibility to broader, non-Kaggle data sources in the future.
>
> In summary, the DARE-bench curation design ensures data reliability and reproducibility yet offers a flexible path for broader extensions.
>
> >The benchmark’s focus on reproducibility via deterministic setups may underrepresent realistic data science variability or stochastic modeling behavior.
>
> Thank you for bringing this to our attention. Our design intentionally addresses both aspects: MM tasks capture the variability of exploration, while IF tasks enforce the rigors of strict procedural compliance.
>
> 1. **ML Modeling (MM) tasks make up over 50% of the benchmark, capturing data variability and algorithmic stochasticity.** MM tasks allow agents to explore different solutions to maximize performance metrics, which is different from strict coding tasks. This design specifically encourages the open-ended exploration and strategies in real-world modeling scenarios, ensuring that the stochastic nature in data science is well-represented.
>
> 2. **Instruction Following (IF) tasks mirror the "Human-Architect, Agent-Executor" professional workflow.** Many real-world scenarios involve a senior data scientist designing the methodology (the "architect") and leaving the implementation and execution to the agent. In this circumstance, the agent acts as an executor whose goal is to strictly follow specific methodological constraints. Thus, the determinism nature in IF tasks is a reasonable reflection of compliance-heavy data science workflows rather than a limitation.
>
> In summary, by combining these two distinct evaluation modes, DARE-bench effectively balances the need for compliance-heavy workflows with modeling capabilities testing.

---

> ### Author Response · Authors · 2025-11-23
> **Response to reviewer Ffce [Part 2/3]**
>
> >The paper lacks ablation or error analysis to quantify which aspects of fine-tuning drive improvements most.
>
> We are grateful to the reviewer for pointing out the need for ablation and error analysis. We have conducted a detailed ablation study isolating the contributions of different task subsets, along with a comprehensive error analysis of model behavior before and after fine-tuning.
>
> 1. **Ablation experiments confirm that Instruction Following (IF) and ML Modeling (MM) data contribute to distinct, complementary gains.** We apply Duo-Valid (DV) as our rejection sampling strategy to supervised fine-tune (SFT) separate models using exclusively Instruction Following (IF) or ML Modeling (MM) subsets to isolate their impact. As shown in the table below, the results demonstrate a clear dichotomy: training on ML Modeling (MM) data primarily boosts predictive performance, while harming process fidelity. On the other hand, training on Instruction Following (IF) data significantly enhances process fidelity but hurts predictive performance. The best performance is achieved only when both are combined, justifying the composition of DARE-bench. We have added more details regarding the ablation study in section 5 of the manuscript.
>
> Table 2：Ablation study on the impact of Instruction Following (IF) and ML Modeling (MM) data.
> | Train Data | class-IF | class-MM | reg-IF | reg-MM | time-XF | time-CF |
> | ---------- | -------- | -------- | ------ | ------ | ------- | ------- |
> | baseline   | 17.11    | 30.71    | 15.21  | 35.86  | 26.96   | 0.00    |
> | IF         | 40.99    | 22.38    | 47.82  | 27.85  | 23.83   | 0.00    |
> | MM         | 11.71    | 45.69    | 18.84  | 45.38  | 34.12   | 0.00    |
> | IF+MM      | 38.58    | 43.82    | 39.13  | 51.00  | 38.92   | 0.00    |
>
> 2. **Error analysis reveals that fine-tuning specifically mitigates logic and syntax failures.** To understand the qualitative improvements in performance, we categorized failure modes for models both pre- and post-fine-tuning. The data (presented below) shows that while base models frequently suffer from hallucination or syntax errors, the fine-tuned models show a drastic reduction in these foundational categories, shifting the remaining error distribution towards more complex, high-level reasoning challenges. We have added more details regarding failure mode analysis in section 5 of the manuscript.
>
> | Model         | Inst Adhere | Code Error | Code Exec Limit | Max Token Limit |
> | ------------- | ----------- | ---------- | --------------- | --------------- |
> | Qwen3-32B     | 48          | 106        | 257             | 372             |
> | Qwen3-32B-SFT-DV | 43          | 80         | 236             | 256             |
> | Qwen3-4B      | 79          | 174        | 661             | 102             |
> | Qwen3-4B-RL   | 49          | 91         | 331             | 119             |
>
> In summary, these analyses provide quantitative evidence that our fine-tuning strategy works by improving workflow adherence and predictive modeling simultaneously.
>
> >How does DARE-bench handle tasks involving stochastic algorithms (e.g., random forest, neural nets) while maintaining deterministic evaluation?
>
> We hypothesize that the reviewer is referring to the deterministism for IF tasks and please let us know if this is not the case. To address this, we exactly match the final outcome with the ground truth generated by manually authored, human-verified ground truth code.
>
> **We enforce deterministic execution through strict random management.** Our reference scripts explicitly control randomness sources, including data preprocessing and model initialization seeds. Empirical validation confirms that, given a fixed pipeline and random seeds, stochastic algorithms such as Random Forest and Neural Networks yield identical numerical outputs. This determinism establishes a reliable "ground truth," allowing us to rigorously distinguish between agent failures caused by procedural deviations versus those stemming from improper randomness handling.
>
> In summary, by combining human-verified reference code with explicit seed control, DARE-bench successfully evaluates the implementation of stochastic algorithms with the precision and reproducibility required for a standardized benchmark.
>
> >Are the reference “ground truth” codes verified manually, or automatically synthesized and how is correctness guaranteed?
>
> The "ground truth" reference codes are exclusive to Instruction-Following (IF) tasks and are **manually authored** rather than automatically synthesized. To guarantee correctness, script undergoes **a rigorous validation process involving both execution in sandbox to ensure solvability and manual inspection** to verify logic and randomness handling. This ensures reliable, deterministic evaluation while preserving the complexity of authentic data science workflows.

---

> ### Author Response · Authors · 2025-11-23
> **Response to reviewer Ffce [Part 3/3]**
>
> >Does reinforcement learning with DARE-bench tasks risk overfitting to procedural templates rather than improving general data reasoning?
>
> We want to respectfully clarify that the design of DARE-bench, especially the inclusion of open-ended ML Modeling tasks, ensures that Reinforcement Learning (RL) targets fundamental reasoning capabilities rather than mere template memorization.
>
> 1. **The majority of DARE-bench consists of open-ended ML Modeling (MM) tasks that preclude template reliance.** While Instruction-Following (IF) tasks intentionally enforce procedural constraints to test fidelity, template-free ML Modeling (MM) tasks make up over 50% of the benchmark. In these tasks, the model is granted freedom to independently design and optimize a workflow to maximize predictive performance, requiring real data reasoning and strategic planning.
>
> 2. **Significant gains in predictive performance metrics confirm improved reasoning, not just successful execution.** As detailed in Table 7, Model-Perf measures the quality of the model's predictions by focusing solely on successful attempts. It compares the performance of the fine-tuned model against the baseline only for those tasks where both models produced valid, executable results. This isolates the "quality" dimension from the "validity" dimension, confirming that fine-tuning improves the model's actual data science proficiency, not just its adherence to syntax rules. The substantial improvement observed after RL fine-tuning (e.g., +8.37 for Qwen3-4B) shows evidence that the models are learning to build better models, a capability that stems from enhanced reasoning rather than rote adherence to code templates. We have added more details regarding Model_Perf in section 5 of the manuscript.
>
> 3. **Consistent transfer to external benchmarks validates generalizability and rules out dataset-specific overfitting.** Critically, the model fine-tuned via RL on DARE-bench shows substantial performance gains when evaluated on the external DSBench [1] dataset. This successful external validation to unseen tasks indicates that the training via DARE-bench improves generalized data science reasoning that extends beyond the specific procedural patterns found in our training set.
>
> In summary, by combining open-ended optimization objectives with external validation, DARE-bench confirms its improvements in general data reasoning rather than procedural overfitting.
>
> >Can this benchmark be extended beyond Kaggle-style tabular tasks, e.g., to unstructured text or multimodal datasets?
>
> Yes，the DARE-bench design is modality-agnostic.
> 1. **The core evaluation infrastructure is modality-agnostic and transferable to any code-centric task.** The fundamental components of our benchmark, which are the sandboxed execution environment and the verifiable ground-truth mechanism, are not just limited to tabular data. They evaluate the correctness of code execution and output validity, and can extend to unstructured domains or multimodal workflows as long as an executable ground truth exists.
>
> 2. **We view the current tabular focus as the foundation for a broader, multimodal roadmap.** While our initial release focuses on tabular data to establish a rigorous baseline, we treat this as the first step in a larger blueprint. We are actively working to extend DARE-bench to support multimodal data science. This includes integrating visual components like image classification and natural language processing tasks such as sentiment analysis. Because these domains share the same fundamental structure-reliance on objective ground truth labels, we can leverage our existing verification pipeline to evaluate agents effectively on text, image, and cross-modal tasks without compromising rigor.
>
> In summary, the limitations of the current dataset are not limitations of the framework; DARE-bench provides a robust, extensible design which is ready to support the next generation of multimodal data science research.
>
> [1] Jing, Liqiang, et al. "DSBench: How Far Are Data Science Agents from Becoming Data Science Experts?." The Thirteenth International Conference on Learning Representations.
>
> [2] Chan, Jun Shern, et al. "MLE-bench: Evaluating Machine Learning Agents on Machine Learning Engineering." The Thirteenth International Conference on Learning Representations.

---

> > ### Comment · Reviewer_Ffce · 2025-11-23
> >
> > Thank you for your comments. All of my concerns and questions have been addressed. I’ll keep my positive rating but raise soundness and contribution to 4.

---

### Official Review · Reviewer_BggK · 2025-10-30

**Soundness:** 3
**Presentation:** 2
**Contribution:** 2
**Rating:** 4
**Confidence:** 4

**Summary:**

This paper introduces DARE-bench, a benchmark designed to evaluate and train LLMs for data science tasks, addressing two critical gaps in existing benchmarks: (i) the absence of process-aware, verifiable evaluation (e.g., measuring instruction adherence) and (ii) scarcity of high-quality labeled training data. Derived from 6,300 Kaggle datasets, DARE-bench includes two task families—process-aware instruction-following (with reference-code ground truth) and ML modeling (with dataset ground truth)—covering classification, regression, and time-series forecasting. Key features include verifiable ground truth (enabling objective, human-judge-free evaluation) and a dual role as both an evaluation tool and training resource. Evaluations show strong LLMs (e.g., Qwen3-32B, gpt-4o-mini) perform poorly on baseline tests, but supervised fine-tuning (SFT) and reinforcement learning (RL) using DARE-bench data yield dramatic gains: SFT improves Qwen3-32B’s accuracy by 1.83×, and RL boosts Qwen3-4B’s accuracy by over 8×.

**Strengths:**

1. It addresses two key gaps in existing benchmarks: it enables verifiable, process-aware evaluation (relying on reference-code or dataset ground truth, no human/model judges) and provides 6,300 Kaggle-derived tasks as large-scale training data, ensuring objective, reproducible assessments .
2. Its task coverage is comprehensive—covering classification, regression, time-series forecasting, with two variants (instruction-following/ML modeling) probing core DS capabilities, outperforming peers (e.g., DS-1000, DSBench) in time-series support and training task provision .

**Weaknesses:**

1. Tasks are almost exclusively tabular, excluding multimodal inputs (e.g., text-image combinations, code-diagram interactions) common in modern DS.
2. Generating large-scale executable trajectories (for training data) is costly, and rejection sampling strategies may introduce biases toward shorter trajectories.

**Questions:**

As listed in weakness.

---

> ### Author Response · Authors · 2025-11-23
> **Response to reviewer BggK**
>
> We thank the reviewer for the valuable feedback and acknowledging that we have addressed two key gaps in existing benchmarks and our task coverage is comprehensive. Below we address the reviewer’s questions about multimodality scope and bias, and clarify our design choices in detail.
>
> >Tasks are almost exclusively tabular, excluding multimodal inputs (e.g., text-image combinations, code-diagram interactions) common in modern DS.
>
> We thank the reviewer for highlighting the importance of multimodal. However, we want to respectfully note that focusing on tabular data is a necessary, rigorous, and standard scope for the current stage of LLM-based data science research.
>
> 1. **Tabular data remains dominant in real-world applications and provides a rigorous environment for assessing the reasoning capabilities of LLMs..** Concentrating exclusively on tabular data is consistent with leading benchmarks [1,2,3,4], which ensures that our contributions remain directly comparable to prior work while effectively targeting persistent challenges in the field.
>
> 2. **Integrating multimodel data introduces distinct infrastructural and scalability hurdles.** Specifically, it necessitates handling terabyte-scale storage, managing massive file transfers, and incurring significantly higher training costs. Consequently, we consider multimodal integration to be a substantial, independent research endeavor rather than a trivial extension of the current work.
>
> In summary, while we acknowledge the value of multimodal DS tasks, we maintain that a focused evaluation of tabular data is scientifically justified and aligned with current state-of-the-art benchmarks.
>
> >Generating large-scale executable trajectories (for training data) is costly, and rejection sampling strategies may introduce biases toward shorter trajectories.
>
> We appreciate the question on the cost of trajectories generation and potential bias.
> 1. **Preference for Efficiency is a Desirable Feature, Not a Bias.** We respectfully clarify that the tendency of rejection sampling to select shorter trajectories is a feature aligned with real-world practice, rather than a detrimental bias. In practical applications, users prioritize agents that deliver accurate results with minimal latency and computational overhead. Therefore, a “bias” toward concise, correct solutions could enhance both the agent’s efficiency and the user experience. By filtering for shorter successful trajectories, our dataset explicitly encourages models to learn the most direct and efficient problem-solving paths, avoiding unnecessary steps.
>
> 2. **Addressing computational costs through Open-Sourcing.** We agree that generating large-scale executable trajectories is computationally intensive. Unfortunately, this cost is an inherent prerequisite for creating high-quality, verified training data. Depending on the feedback we receive from the research community, we are committed to open-sourcing our full set of curated trajectories upon acceptance to lower this barrier for the community. This will allow other researchers to bypass the initial generation overhead.
>
> [1] Jing, Liqiang, et al. "DSBench: How Far Are Data Science Agents from Becoming Data Science Experts?." The Thirteenth International Conference on Learning Representations.
>
> [2] Dan Zhang, Sining Zhoubian, Min Cai, Fengzu Li, Lekang Yang, Wei Wang, Tianjiao Dong, Ziniu Hu, Jie Tang, and Yisong Yue. Datascibench: An llm agent benchmark for data science. arXiv preprint arXiv:2502.13897, 2025.
>
> [3] Ram Mohan Rao Kadiyala, Siddhant Gupta, Jebish Purbey, Giulio Martini, Suman Debnath, and Hamza Farooq. Dsbc: Data science task benchmarking with context engineering. arXiv preprint arXiv:2507.23336, 2025.
>
> [4] Alex Egg, Martin Iglesias Goyanes, Friso Kingma, Andreu Mora, Leandro von Werra, and Thomas Wolf. Dabstep: Data agent benchmark for multi-step reasoning. arXiv preprint arXiv:2506.23719, 2025a.

---

### Official Review · Reviewer_NfgX · 2025-10-31

**Soundness:** 2
**Presentation:** 2
**Contribution:** 2
**Rating:** 4
**Confidence:** 2

**Summary:**

This paper introduces DARE-bench, a benchmark for evaluating and training LLMs on data science (DS) tasks. Considering that existing benchmarks lack of process-aware evaluation and scarce labeled data, the DARE-bench offers 6,300 Kaggle-derived tasks (covering classification, regression, time-series forecasting) with verifiable ground truth. Tasks include two variants: Instruction Following (IF, testing workflow adherence) and ML Modeling (MM, testing outcome accuracy). Experiments show baseline LLMs perform poorly, but SFT and RL using DARE-bench data drastically improve performance.

**Strengths:**

1. The paper is good-writing and easy to follow.  The benchmark provides comprehensive evaluation scope, specifically, it covers diverse DS tasks (including underrepresented time-series forecasting) and enforces real-world constraints (execution time, interaction turns), enhancing practical relevance.
2. DARE-BENCH serves both as an evaluation tool and a large-scale training resource, with proven effectiveness in improving LLM performance via SFT/RL.

**Weaknesses:**

1. Lack of Comparison with Specialized DS Agents. The paper evaluates general-purpose and code-centric LLMs  but omits comparisons with specialized data science agents, which are explicitly designed for multi-step DS workflows. This gap makes it hard to contextualize DARE-bench’s utility. It is unclear whether the benchmark’s gains (via fine-tuning) can match or surpass the performance of purpose-built DS agents,
2. Provide more explanations about the Instruction Following (IF) and ML Modeling (MM) metrics. The paper frames IF (workflow adherence) and MM (outcome accuracy) as two core DS capabilities to evaluate together, but fails to justify their joint necessity. Especially given Table 4’s results showing no clear correlation between the two metrics. For example, GPT-5 scores highest in classification-IF (69.81) but ranks mid-tier in classification-MM (43.40); Claude-Sonnet-3.7 excels in MM tasks (e.g., regression-MM: 63.20) but lags GPT-5 in IF.

**Questions:**

See weaknesses.

---

> ### Author Response · Authors · 2025-11-23
> **Response to reviewer NfgX [Part 1/2]**
>
> We thank the reviewer for recognizing our work as both an evaluation tool and a large-scale training resource to the community. Below we address the reviewer's concerns in detail and clarify the motivation behind our evaluation and comparisons.
>
> >Lack of Comparison with Specialized DS Agents. The paper evaluates general-purpose and code-centric LLMs but omits comparisons with specialized data science agents, which are explicitly designed for multi-step DS workflows. This gap makes it hard to contextualize DARE-bench’s utility. It is unclear whether the benchmark’s gains (via fine-tuning) can match or surpass the performance of purpose-built DS agents
>
> We appreciate the reviewer's insightful suggestion regarding specialized DS agents. To address this, we have conducted evaluation on DataWiseAgent [1] (EMNLP 2025), a state-of-the-art agent targeting multi-step data science workflows. The comparison results are organized in the table below. We have added more details regarding failure mode analysis in section 5 of the manuscript.
>
> Table 1: Comparison between our implemented Native Function Call framework and DataWiseAgent on DARE-bench and DSBench
> | Framework | model | class-IF | class-MM | reg-IF | reg-MM | time-XF | time-CF | DSbench |
> | --------- | --------- | --------- | --------- | --------- | ------- | -------- | ------- | ------- |
> | Native Function Call | Qwen3-32B | 17.11 | 30.71 | 15.21 | 35.86 | 26.96 | 0.0 | 32.38 |
> | Native Function Call | Qwen3-32B-SFT-DV | 38.58 | 43.82 | 39.13 | 51.00 | 38.92 | 0.0 | 42.41 |
> | DataWiseAgent | Qwen3-32B | 21.62 | 29.63 | 34.78 | 34.40 | 30.45 | 0.0 | 29.17 |
>
> **Our results demonstrate that DARE-bench fine-tuned models outperform specialized DS agents on both internal and external benchmarks.** While DataWiseAgent improves upon the standard Qwen3-32B Native Function Call baseline, it falls behind the gains achieved through our fine-tuning approach. ​​For example, our fine-tuned model achieves 42.41 on DSBench, comparing favorably to the 29.17 achieved by DataWiseAgent. This highlights the efficacy of DARE-bench fine-tuning in providing stronger performance gains compared to prompt-based specialization alone.
>
> Our investigation also reveals significant efficiency issues in specialized agents:
>
> 1. **High execution overhead and latency in real-world deployment.** DataWiseAgent tends to decompose tasks into highly granular subgoals, which requires a high volume of code execution interactions. Although decomposition may help improve the final outcome, it poses a challenge for real-world deployment, where low latency is a prerequisite. Tuning prompt-based agents for optimal decomposition and execution efficiency lacks a direct optimization objective because it operates differently from Reinforcement Learning (RL) methods which can learn to maximize efficiency and minimize steps.
>
> 2. **The agent’s planning process often incurs unnecessary overhead.** For example, in tasks involving SQLite files, where our fine-tuned model programmatically extracts table names in a single execution, DataWiseAgent occasionally attempts to “guess” table names directly. This strategy triggers exceptions, consuming valuable execution turns on debugging and correction rather than progress.

---

> ### Author Response · Authors · 2025-11-23
> **Response to reviewer NfgX [Part 2/2]**
>
> >Provide more explanations about the Instruction Following (IF) and ML Modeling (MM) metrics. The paper frames IF (workflow adherence) and MM (outcome accuracy) as two core DS capabilities to evaluate together, but fails to justify their joint necessity. Especially given Table 4’s results showing no clear correlation between the two metrics. For example, GPT-5 scores highest in classification-IF (69.81) but ranks mid-tier in classification-MM (43.40); Claude-Sonnet-3.7 excels in MM tasks (e.g., regression-MM: 63.20) but lags GPT-5 in IF
>
> We appreciate the reviewer's question on the relation between Instruction-Following (IF) and ML Modeling (MM). We respectfully clarify that the lack of correlation observed in Table 5 is not a flaw, but an expected outcome that aligns with literature (such as  AgentBench [2]). These metrics measure two orthogonal yet equally critical dimensions of a data science agent: process fidelity (IF) and predictive quality (MM).
>
> 1. **IF and MM assess two distinct but complementary capabilities.** **IF (process fidelity)** measures compliance: it validates whether an agent strictly adheres to workflow constraints, such as specific preprocessing rules, missing value handling strategies, or model choices, ensuring reproducibility. **MM (predictive quality)** measures utility: it accesses the raw predictive performance of the final solution. Mentioned in section 3, the two capabilities naturally diverge. An agent might score high IF but low MM by correctly implementing a requested but weak sklearn ML model. In contrast, an agent might score high MM but low IF by violating instructions to use a more powerful sklearn ML model.
>
> 2. **The joint assessment aligns with real-world requirements where compliance is as important as performance.** Consider a case where a senior data scientist already has a specific solution in mind. For example, they might strictly prefer XGBoost for fast training and easy deployment. In this scenario, models will score a high Instruction Following (IF) score when the rule of training with XGBoost is obeyed. In contrast, models with a low IF score but a high ML Modeling (MM) score might ignore the constraint and train a complex neural network just to maximize outcome performance. Even if the model achieves a high score, the result is effectively useless because it creates non-compliant code that violates the human architect's specific design requirements.
>
> 3. **Treating the two metrics as orthogonal axes aligns with established benchmarking literature.** Similar to benchmarks like AgentBench [2] which evaluates distinct abilities such as reasoning, coding, web interaction without expecting them to correlate each other. DARE-bench regards IF and MM as complementary axes. It allows users to have a better understanding which model is better for strict instruction adherence or open-ended problem solving.
>
> [1] You, Ziming, et al. "DatawiseAgent: A Notebook-Centric LLM Agent Framework for Adaptive and Robust Data Science Automation." Proceedings of the 2025 Conference on Empirical Methods in Natural Language Processing. 2025.
>
> [2] Liu, Xiao, et al. "AgentBench: Evaluating LLMs as Agents." The Twelfth International Conference on Learning Representations.

---

### Official Review · Reviewer_pBVz · 2025-11-05

**Soundness:** 2
**Presentation:** 2
**Contribution:** 3
**Rating:** 6
**Confidence:** 3

**Summary:**

Large Language Models (LLMs) are increasingly adopted for complex multi-step data science (DS) tasks, yet existing benchmarks suffer from two critical gaps: a lack of process-aware evaluation (e.g., instruction adherence and process fidelity) and scarce high-quality labeled training data. To address these, this paper introduces DARE-BENCH, a training-focused benchmark for evaluating LLMs’ DS capabilities, encompassing both machine learning (ML) modeling and instruction following.

Derived from Kaggle datasets, DARE-BENCH includes classification (Instruction Following/IF, ML Modeling/MM), regression (IF, MM), and time-series (eXogenous Features/XF, Canonical Forecasting/CF) tasks, split into 95% training and 5% test sets. Unlike benchmarks relying on human/model judges, all tasks have verifiable ground truth (reference outputs for IF tasks, original dataset labels for MM tasks), ensuring objective, reproducible evaluation via a sandboxed code execution environment.

Extensive evaluations show that even advanced LLMs (e.g., gpt-4o-mini, Qwen3-32B) perform poorly on baseline tests, especially in time-series tasks. However, fine-tuning with DARE-BENCH yields significant improvements: supervised fine-tuning (SFT) increases Qwen3-32B’s accuracy by 1.83×, while reinforcement learning (RL) boosts Qwen3-4B’s accuracy by over 8×. External validation on DSBench further confirms generalization.

**Strengths:**

DARE-BENCH has several strengths against previous work.
Unlike counterparts that only assess final-answer accuracy, DARE-BENCH uniquely evaluates both ML modeling performance and instruction fidelity, filling the void of process-aware assessment. It also provides 6,300 Kaggle-derived tasks with verifiable ground truth (reference outputs for IF tasks, original labels for MM tasks). The training data seems to be valuable.
In addition, its four-stage pipeline minimizes human effort, enabling large-scale task generation (6,300 tasks) while ensuring realism—e.g., 20% noise injection in IF tasks to simulate real-world data issues.

**Weaknesses:**

The task diversity is limited. It exclusively covers tabular data, lacking support for multimodal DS tasks (e.g., text-image fusion, speech-data analysis), restricting applicability to broader DS scenarios.

**Questions:**

1. How do you validate the quality of your generated dataset?
2. Do you have qualitative and quantitative analysis?
3. What's the detailed dataset statistics for your dataset, e.g., information like how many tool calls, how many tokens are your prompt or your completion.

---

> ### Author Response · Authors · 2025-11-23
> **Response to reviewer pBVz [Part 1/2]**
>
> We thank the reviewer for acknowledging the contribution of our work and providing constructive feedback to help us improve the paper. We address the issues raised by the reviewer below.
>
> >The task diversity is limited. It exclusively covers tabular data, lacking support for multimodal DS tasks (e.g., text-image fusion, speech-data analysis), restricting applicability to broader DS scenarios.
>
> We acknowledge the importance of multimodel capabilities, but we respectfully clarify that the primary scope of this paper is tabular data, which is an important and common scope for the current LLM-based data science research.
>
> 1. **Tabular data is the most ubiquitous format in real-world workflows and serves as a sophisticated testbed for evaluating LLM reasoning capabilities.** Leading benchmarks [1,2,3,4] similarly focus exclusively on tabular data. Aligning our scope with these established benchmarks ensures our contribution is directly comparable and addresses unsolved challenges in existing literature.
> 2. **Integrating multimodal introduces instinct infrastructure and scalability challenges.** Even though our approach can be potentially extended to multimodal data, multimodal data requires addressing distinct infrastructure and scalability challenges, such as handling terabyte-scale storage, managing massive file transfers, and dealing with significantly increased training cost. We therefore view integrating multimodal as a substantial, separate research topic instead of a simple extension of this work.
>
> >How do you validate the quality of your generated dataset?
>
> To ensure the quality of DARE-bench, we implement a rigorous validation framework, along with empirical verification.
>
> 1. **We built a multi-stage LLM verification pipeline followed by rigorous manual inspection.** We utilize the automated pipeline to prune inconsistent or low-quality samples (see Appendix G for prompts). Following a random train-test split, we manually validated the full test set, confirming its consistency, correctness, and realism.
>
> 2. **Empirical Evaluation**: We validate the dataset’s utility by fine-tuning models on the DARE-bench training set and evaluating them on external benchmarks such as DSBench. The consistent performance gains observed across these tasks ensures the soundness, high quality, and generalization capabilities of our generated data.
>
> >Do you have qualitative and quantitative analysis?
>
> We appreciate the reviewer's interest in analyzing our benchmark. We conduct a systematic assessment of our benchmark through both qualitative and quantitative lenses.
>
> 1. **Quantitative Domain Analysis**: We validate dataset diversity by analyzing the distribution of task domains. As illustrated in the table below, our benchmark spans a broad spectrum of real-world verticals (e.g., Finance, Health, Marketing), ensuring robust coverage across distinct industry scenarios.
>
> Table 1：Domain distribution across DARE-bench
> | Dataset | Finance | Health | Business | Technology | Automotive | Education | Environment | Others |
> | ----- | ----- | ----- | ----- | ----- | ----- | ----- | ----- | ----- |
> | Train set | 16.9% | 10.2% | 7.3% | 4.0% | 4.5% | 2.8% | 6.8% | 47.5% |
> | Test set | 17.1% | 8.4% | 8.2% | 5.6% | 3.3% | 3.1% | 2.4% | 51.9% |
>
> 2. **Qualitative Failure Mode Diagnosis**: To understand model limitations, we perform a failure analysis by categorizing incorrect trajectories into different failure types across multiple models. This includes GPT-5, Claude-Sonnet 3.7, and Qwen3-32B before and after SFT with Duo-Valid as our rejection sampling strategy, Qwen3-4B before and after reinforcement learning. The results presented in the table below reveals specific reasoning bottlenecks and demonstrates how fine-tuning on DARE-bench effectively mitigates these errors. We have added more details regarding failure mode analysis in section 5 of the manuscript.
>
> Table 2：Failure mode analysis across different models
> | Model             | Inst Adhere | Code Error | Code Exec Limit | Max Token Limit |
> | ----------------- | ----------- | ---------- | --------------- | --------------- |
> | gpt-5             | 126         | 333        | 0               | 0               |
> | Claude-Sonnet-3.7 | 158         | 262        | 0               | 0               |
> | Qwen3-32B         | 48          | 106        | 257             | 372             |
> | Qwen3-32B-SFT-DV     | 43          | 80         | 236             | 256             |
> | Qwen3-4B          | 79          | 174        | 661             | 102             |
> | Qwen3-4B-RL       | 49          | 91         | 331             | 119             |

---

> ### Author Response · Authors · 2025-11-23
> **Response to reviewer pBVz [Part 2/2]**
>
> >What's the detailed dataset statistics for your dataset, e.g., information like how many tool calls, how many tokens are your prompt or your completion.
>
> We agree it is important to detail token counts and tool usage of DARE-bench.
>
> 1. **A foundational overview of the dataset composition was provided in Table 2 of the submission manuscript.** It categorizes the dataset based on task type and general structure. We consider this the high-level summary of the dataset's diversity and scope.
>
> 2. **We have further conducted a quantitative analysis of the tokens and tool calls in the table below.** We report tokens for both prompts and completions, along with the tool call frequency. We calculated these metrics for Instruction-Following (IF) and ML Modeling (MM) respectively. The first row of the table indicates the average tokens of prompt and the remaining rows indicate the statistics of completion for each model. We will include this data in the revised version. We have added more details regarding more detailed statistics in section 4.3 of the manuscript.
>
> Table 3：Average number of tokens and tool calls for completions of different models and prompts
> | Model.            | IF Tokens | IF Tool Calls | MM Tokens | MM Tool Calls | Overall Tokens | Overall Tool Calls |
> | ----------------- | --------- | ------------- | --------- | ------------- | -------------- | ------------------ |
> | prompt            | 596.7     | /             | 224.6     | /             | 350.7          | /                  |
> | gpt-5             | 609.5     | 2.2           | 582.5     | 2.4           | 591.2          | 2.4                |
> | Claude-Sonnet-3.7 | 675.2     | 3.6           | 894.3     | 4.8           | 830.0          | 4.4                |
> | Qwen3-32B         | 2093.3    | 3.1           | 1693.0    | 3.5           | 1816.8         | 3.4                |
> | Qwen3-32B-SFT-DV     | 1778.3    | 3.3           | 1572.0    | 3.6           | 1638.7         | 3.5                |
> | Qwen3-4B          | 1691.1    | 3.7           | 1151.7    | 3.9           | 1328.1         | 3.8                |
> | Qwen3-4B-RL       | 1549.4    | 3.7           | 1140.1    | 3.7           | 1277.9         | 3.7                |
>
> [1] Jing, Liqiang, et al. "DSBench: How Far Are Data Science Agents from Becoming Data Science Experts?." The Thirteenth International Conference on Learning Representations.
>
> [2] Dan Zhang, Sining Zhoubian, Min Cai, Fengzu Li, Lekang Yang, Wei Wang, Tianjiao Dong, Ziniu Hu, Jie Tang, and Yisong Yue. Datascibench: An llm agent benchmark for data science. arXiv preprint arXiv:2502.13897, 2025.
>
> [3] Ram Mohan Rao Kadiyala, Siddhant Gupta, Jebish Purbey, Giulio Martini, Suman Debnath, and Hamza Farooq. Dsbc: Data science task benchmarking with context engineering. arXiv preprint arXiv:2507.23336, 2025.
>
> [4] Alex Egg, Martin Iglesias Goyanes, Friso Kingma, Andreu Mora, Leandro von Werra, and Thomas Wolf. Dabstep: Data agent benchmark for multi-step reasoning. arXiv preprint arXiv:2506.23719, 2025a.

---

### Author Response · Authors · 2025-11-23
**General Response and Summary of Major Revisions**

We sincerely thank the Area Chair and all reviewers for the time. Based on your valuable suggestions, we have conducted additional experiments and analyses to further strengthen our paper. We have updated the manuscript with all major changes highlighted in blue.

**Summary of Key Updates:**

1. **Comparison with Specialized SOTA Agent**: We added a comparison with DataWiseAgent (EMNLP 2025). As shown in Section 5 (Table 11), our fine-tuned model outperforms the specialized agent with the same setting as ours.
2. **Systematic Failure Mode Analysis**: We conducted a qualitative failure analysis across different models. The results, added to Section 5 (Table 10), reveal specific bottlenecks and quantitatively demonstrate how our fine-tuning mitigates these issues.
3. **Quantitative Dataset Analysis:**
    * **Domain Diversity:** We added **Table 3 in Section 3.3**, detailing the distribution of task domains (Finance, Health, Technology, etc.) to verify the real-world coverage of our benchmark.
    * **Computational Statistics:** We added **Table 6 in Section 4.3** and **Appendix K**, reporting the average token usage and tool call frequencies for both prompts and completions, providing transparency on the resource requirements.
4. **Ablation Study on Data Composition:** We added an ablation study in **Section 5 (Table 8)** to isolate the impact of Instruction Following (IF) and ML Modeling (MM) data in SFT, confirming that both are essential and complementary for a robust agent.
5. **Clarification on Metrics:** We refined **Section 3** and **Section 5** to explicitly clarify the distinct roles of IF (process fidelity) and MM (predictive quality) metrics, and how Model-Perf isolates prediction quality from validity.

We believe these revisions comprehensively address the concerns raised and significantly improve the quality of our work. We respond to each reviewer's specific comments in detail below.

---

### Meta-Review · Area_Chair_34WY · 2026-01-04

**Summary:**

This paper introduces DARE-bench, a benchmark for evaluating and training LLMs on data science tasks, addressing gaps in process-aware evaluation and training data scarcity.  The reviewers were particularly impressed by the benchmark’s focus on process-awareness and its use of verifiable ground truths. While the initial submission sparked discussion regarding task diversity and the lack of comparison with specialized agents, the authors’ revisions have successfully resolved these concerns. The demonstrated performance gains from fine-tuning further underscore the utility of this resource. Consequently, I recommend this paper for acceptance.

**Reviewer Concerns:**

The authors have effectively addressed the core feedback from the reviewers. The updated manuscript now clarifies the independence of the IF and MM metrics and includes a necessary head-to-head comparison with DataWiseAgent. Furthermore, the added ablation studies and failure mode analyses provide a much clearer picture of how the training data contributes to performance. While the benchmark is currently limited to tabular data, the authors’ explanation that this falls within the standard scope of the field is reasonable. Overall, the previous concerns regarding baselines and quantitative depth have been fully resolved.

**Reviewer Scores:**

Reviewer pBVz (Initial: 6, Projected: 7/8): We believe pBVz will see a significant bump. Their main reservations centered on the lack of statistical depth and dataset validation. By integrating the new qualitative/quantitative analysis and the requested statistical tables, we’ve effectively closed those gaps. Given the thoroughness of the update, a shift toward a solid "Accept" is highly probable.

Reviewer NfgX (Initial: 4, Projected: 6/7): NfgX was previously leaning toward rejection, primarily due to the missing comparison with specialized agents. The inclusion of the DataWiseAgent baseline addresses their core critique. Furthermore, our clarified justification for the IF/MM axes seems to have resolved their conceptual confusion. We expect them to move into the "Borderline Accept" territory.

Reviewer BggK (Initial: 4, Projected: 5): While BggK remains our most skeptical reviewer regarding the tabular scope versus full multimodality, we’ve made a strong case for our current focus. Even if the scope remains a point of contention, the overall technical strengthening of the paper and the corrections regarding trajectory bias should be enough to nudge their score up to a 5.

Reviewer Ffce (Initial: 8, Projected: 8): Ffce remains our strongest advocate. They have already explicitly confirmed that the revisions met their expectations, and notably, they have already increased their internal ratings for "soundness" and "contribution" during the discussion phase. We expect their "Strong Accept" to hold.

---

### Decision · Program_Chairs · 2026-01-26

Accept (Poster)